# SIMPLICIAL HOPFIELD NETWORKS

**Thomas F. Burns**
Neural Coding and Brain Computing Unit
OIST Graduate University, Okinawa, Japan
`thomas.burns@oist.jp`

**Tomoki Fukai**
Neural Coding and Brain Computing Unit
OIST Graduate University, Okinawa, Japan
`tomoki.fukai@oist.jp`

## ABSTRACT

Hopfield networks are artificial neural networks which store memory patterns on the states of their neurons by choosing recurrent connection weights and update rules such that the energy landscape of the network forms attractors around the memories. How many stable, sufficiently-attracting memory patterns can we store in such a network using $N$ neurons? The answer depends on the choice of weights and update rule. Inspired by setwise connectivity in biology, we extend Hopfield networks by adding setwise connections and embedding these connections in a simplicial complex. Simplicial complexes are higher dimensional analogues of graphs which naturally represent collections of pairwise and setwise relationships. We show that our simplicial Hopfield networks increase memory storage capacity. Surprisingly, even when connections are limited to a small random subset of equivalent size to an all-pairwise network, our networks still outperform their pairwise counterparts. Such scenarios include non-trivial simplicial topology. We also test analogous modern continuous Hopfield networks, offering a potentially promising avenue for improving the attention mechanism in Transformer models.

## 1 INTRODUCTION

Hopfield networks (Hopfield, 1982)[1] store memory patterns in the weights of connections between neurons. In the case of pairwise connections, these weights translate to the synaptic strength between pairs of neurons in biological neural networks. In such a Hopfield network with $N$ neurons, there will be $\binom{N}{2}$ of these pairwise connections, forming a complete graph. Each edge is weighted by a procedure which considers $P$ memory patterns and which, based on these patterns, seeks to minimise a defined energy function such that the network's dynamics are attracted to and ideally exactly settles in the memory pattern which is nearest to the current states of the neurons. The network therefore acts as a content addressable memory – given a partial or noise-corrupted memory, the network can update its states through recurrent dynamics to retrieve the full memory. Since its introduction, the Hopfield network has been extended and studied widely by neuroscientists (Griniasty et al., 1993; Schneidman et al., 2006; Sridhar et al., 2021; Burns et al., 2022), physicists (Amit et al., 1985; Agliari et al., 2013; Leonetti et al., 2021), and computer scientists (Widrich et al., 2020; Millidge et al., 2022). Of particular interest to the machine learning community is the recent development of modern Hopfield networks (Krotov & Hopfield, 2016) and their close correspondence (Ramsauer et al., 2021) to the attention mechanism of Transformers (Vaswani et al., 2017).

An early (Amit et al., 1985; McEliece et al., 1987) and ongoing (Hillar & Tran, 2018) theme in the study of Hopfield networks has been their memory storage capacity, i.e., determining the number of memory patterns which can be reliably stored and later recalled via the dynamics. As discussed in Appendix A.1, this theoretical and computational exercise serves two purposes: (i) improving the memory capacity of such models for theoretical purposes and computational applications; and (ii) gaining an abstract understanding of neurobiological mechanisms and their implications for biological memory systems. Traditional Hopfield networks with binary neuron states, in the limit of $N \to \infty$ and $P \to \infty$, maintain associative memories for up to approximately $0.14N$ patterns (Amit et al.,

---

[1]After the proposal of Marr (1971), many similar models of associative memory were proposed, e.g., those of Nakano (1972), Amari (1972), Little (1974), and Stanley (1976) – all before Hopfield (1982). Nevertheless, much of the research literature refers to and seems more proximally inspired by Hopfield (1982). Many of these models can also be considered instances of the Lenz-Ising model (Brush, 1967) with infinite-range interactions.

1985; McEliece et al., 1987), and fewer if the patterns are statistically or spatially correlated (Löwe, 1998). However, by a clever reformulation of the update rule based on the network energy, this capacity can be improved to $N^{d-1}$, where $d \geq 2$ (Krotov & Hopfield, 2016), and even further to $2^{N/2}$ (Demircigil et al., 2017). Networks using these types of energy-based update rules are called modern Hopfield networks. Krotov & Hopfield (2016) (like Hopfield (1984)) also investigated neurons which took on continuous states. Upon generalising this model by using the softmax activation function, Ramsauer et al. (2021) showed a connection to the attention mechanism of Transformers (Vaswani et al., 2017). However, to the best of our knowledge, these modern Hopfield networks have not been extended further to include explicit setwise connections between neurons, as has been studied and shown to improve memory capacity in traditional Hopfield networks (Peretto & Niez, 1986; Lee et al., 1986; Baldi & Venkatesh, 1987; Newman, 1988). Indeed, Krotov & Hopfield (2016), who introduced modern Hopfield networks, make a mathematical analogy between their energy-based update rule and setwise connections given their energy-based update rule can be interpreted as allowing individual pairs of pre- and post-synaptic neurons to make multiple synapses with each other – making pairwise connections mathematically as strong as equivalently-ordered setwise connections[2]. Demircigil et al. (2017) later proved this analogy to be accurate in terms of theoretical memory capacity. By adding explicit setwise connections to modern Hopfield networks, we essentially allow all connections (pairwise and higher) to increase their strength – following the same interpretation, this can be thought of as allowing both pairwise and setwise connections between all neurons, any of which may be precisely controlled.

Functionally, setwise connections appear in abundance in biological neural networks. What's more, these setwise interactions often modulate and interact with one another in highly complex and nonlinear fashions, adding to their potential computational expressiveness. We discuss these biological mechanisms in Appendix A.2. There are many contemporary models in deep learning which implicitly model particular types of setwise interactions (Jayakumar et al., 2020). To explicitly model such interactions, we have multiple options. For reasons we discuss in Appendix A.3, we choose to model our setwise connections using a simplicial complex.

We therefore develop and study *Simplicial Hopfield Networks*. We weight the simplices of the simplicial complex to store memory patterns and generalise the energy functions and update rules of traditional and modern Hopfield networks. Our main contributions are:

- *We introduce extensions of various Hopfield networks with setwise connections.* In addition to generalising Hopfield networks to include explicit, controllable setwise connections based on an underlying simplicial structure, we also study whether the topological features of the underlying structure influences performance.

- *We prove and discuss higher memory capacity in the general case of simplicial Hopfield networks.* For the fully-connected simplicial Hopfield network, we prove a larger memory capacity than previously shown by Newman (1988); Demircigil et al. (2017) for higher-degree Hopfield networks.

- *We empirically show improved performance under parameter constraints.* By restricting the total number of connections to that of pairwise Hopfield networks with a mixture of pairwise and setwise connections, we show simplicial Hopfield networks retain a surprising amount of improved performance over pairwise networks but with fewer parameters, and are robust to topological variability.

## 2 SIMPLICIAL HOPFIELD NETWORKS

### 2.1 SIMPLICIAL COMPLEXES

Simplicial complexes are mathematical objects which naturally represent collections of setwise relationships. Here we use the combinatorial form, called an *abstract simplicial complex*. Although, to build intuition and visualise the simplicial complex, we also refer to their geometric realisations.

**Definition 2.1.** Let $K$ be a subset of $2^{[N]}$. The subset $K$ is an abstract simplicial complex if for any $\sigma \in K$, the condition $\rho \subseteq \sigma$ gives $\rho \in K$, for any $\rho \subseteq \sigma$.

---

[2]Work by Horn, D. & Usher, M. (1988) study almost the same system but with an slight modification to the traditional update rule, whereas Krotov & Hopfield (2016) use their modern, energy-based update rule.

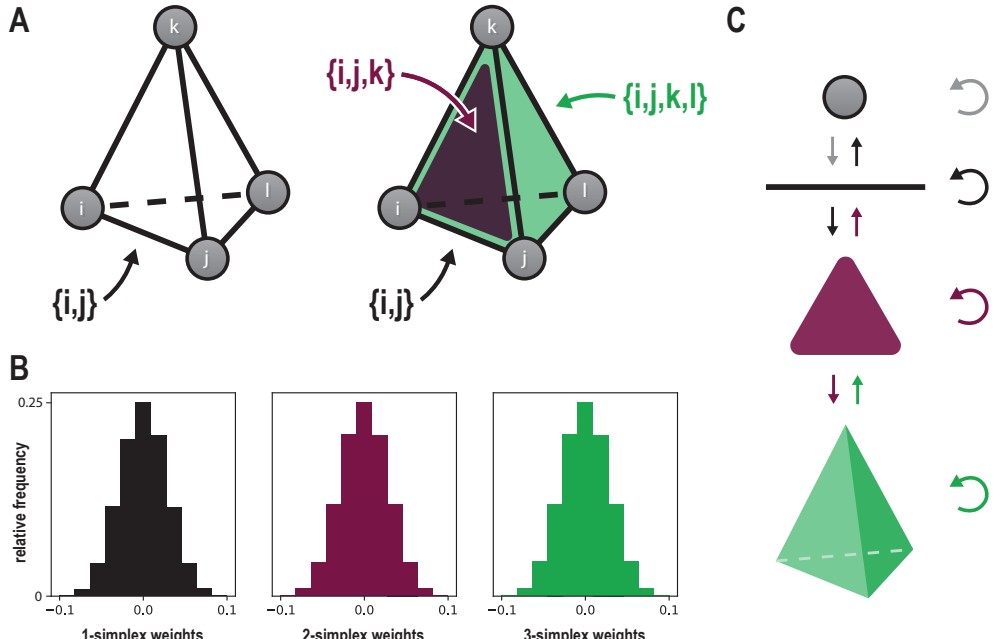

Figure 1: **A.** Comparative illustrations of connections in a pairwise Hopfield network (left) and a simplicial Hopfield network (right) with $N = 4$. In a simplicial Hopfield network, $\sigma = \{i, j\}$ is an edge (1–simplex), $\sigma = \{i, j, k\}$ is a triangle (2–simplex), $\sigma = \{i, j, k, l\}$ is a tetrahedron (3–simplex), and so on. **B.** Connection weight histograms of 1–, 2–, and 3–simplices in a simplicial Hopfield network. In the binary case, the x-axis range is $[-P/N, +P/N]$. Here, $N = 100$ and $P = 10$, thus the range is $[-0.1, +0.1]$. Note that each dimension shows a similar, Gaussian distribution of weights (although there are different absolute numbers of these weights; see 'Mixed diluted networks' in Section 2.2). **C.** Illustration of the hierarchical relationship between elements in the complex, up to 3–simplices, with arrows indicating potential sources of weight modulation or interaction, e.g., between (co)faces or using Hodge Laplacians within the same dimension. Such modulations and interactions (including their biological interpretations) are discussed in Appendices A.2 and A.3.

In other words, an abstract simplicial complex $K$ is a collection of finite sets closed under taking subsets. A member of $K$ is called a *simplex* $\sigma$. A $k$–dimensional simplex (or $k$–simplex) has cardinality $k + 1$ and $k + 1$ faces which are $(k - 1)$–simplices (obtained by omitting one element from $\sigma$). If a simplex $\sigma$ is a face of another simplex $\tau$, we say that $\tau$ is a *coface* of $\sigma$. We denote the set of all $k$–simplices in $K$ as $K_k$.

Geometrically, for $k = 0, 1, 2$, and 3, a $k$–simplex is, respectively, a point, line segment, triangle, and tetrahedron. Therefore, one may visualise a simplicial complex as being constructed by gluing together simplices such that every finite set of vertices in $K$ form the vertices of at most one simplex. This structure makes it possible to associate every setwise relationship uniquely with a $k$–simplex identified by its elements, which in our case are neurons (see Figure 1A). Simplices in $K$ which are contained in no higher dimensional simplices, i.e., they have no cofaces, are called the *facets* of $K$. The dimension of $K$, $\dim(K)$, is the dimension of its largest facet. We call a simplicial complex $K$ a $k$–*skeleton* when all possible faces of dimension $k$ exist and $\dim(K) = k$.

## 2.2 MODEL

A network of $N$ neurons is modelled by $N$ spins. Let $K$ be a simplicial complex on $N$ vertices. In the binary neuron case, $S_j^{(t)} = \pm 1$ at time-step $t$. Given a set of neurons $\sigma$ (which contains the neuron $i$ and is a unique $(|\sigma| - 1)$–simplex in $K$), $w(\sigma)$ is the associated simplicial weight and $S_\sigma^{(t)}$ the product of their spins. Spin configurations correspond to patterns of neural firing, with dynamics

governed by a defined energy. The traditional model is defined by energy and weight functions

$$E = -\sum_{\sigma \in K} w(\sigma) S_\sigma^{(t)} \qquad\qquad w(\sigma) = \frac{1}{N} \sum_{\mu=1}^{P} \xi_\sigma^\mu, \qquad (1)$$

with $\xi_i^\mu (= \pm 1)$ static variables being the $P$ binary memory patterns stored in the simplicial weights. Similar for spins, $\xi_\sigma^\mu$ is the product of the static pattern variables for the set of neurons $\sigma$ in the pattern $\mu$. Figure 1B shows examples of the resulting Gaussian distributions of weights at each dimension of the simplicial complex. We use these weights to update the state of a neuron $i$ by applying the traditional Hopfield update rule

$$S_i^{(t)} = \Theta\left(\sum_{\sigma \in K} w(\sigma) S_{\sigma\setminus i}^{(t-1)}\right) \qquad\qquad \Theta(x) = \begin{cases} 1 & \text{if } x \geq 0 \\ -1 & \text{if } x < 0 \end{cases}. \qquad (2)$$

When $K$ is a 1-skeleton, this becomes the traditional pairwise Hopfield network (Hopfield, 1982). In the modern Hopfield case, the energy function and update rule are

$$E = -\sum_{\mu=1}^{P} \sum_{\sigma \in K} F(\xi_\sigma^\mu S_\sigma^{(t)}) \qquad (3)$$

$$S_i^{(t)} = \text{sgn}\left[\sum_{\mu=1}^{P}\left(F(1 \cdot \xi_i^\mu + \sum_{\sigma \in K} \xi_{\sigma\setminus i}^\mu S_{\sigma\setminus i}^{(t-1)}) - F(-1 \cdot \xi_i^\mu + \sum_{\sigma \in K} \xi_{\sigma\setminus i}^\mu S_{\sigma\setminus i}^{(t-1)})\right)\right], \qquad (4)$$

where the function $F$ can be chosen, for example, to be of a polynomial $F(x) = x^n$ or exponential $F(x) = e^x$ form. When $K$ is a 1-skeleton, this becomes the modern pairwise Hopfield network (Krotov & Hopfield, 2016).

In the continuous modern Hopfield case, spins and patterns take real values $S_j, \xi_j^\mu \in \mathbb{R}$. Patterns are arranged in a matrix $\Xi = (\xi^1, ..., \xi^P)$ and we define the *log-sum-exp function* (lse) for $T^{-1} > 0$ as

$$\text{lse}(T^{-1}, \Xi^T S^{(t)}, K) = T \log\left(\sum_{\mu=1}^{P} \sum_{\sigma \in K} \exp(T^{-1} \Xi_\sigma^\mu S_\sigma^{(t)})\right). \qquad (5)$$

The energy function is

$$E = -\text{lse}(T^{-1}, \Xi^T S^{(t)}, K) + \frac{1}{2} S^{(t)T} S^{(t)}. \qquad (6)$$

For each simplex $\sigma \in K$, we denote the submatrix of the patterns stored on that simplex as $\Xi_\sigma$ (which has dimensions $P \times \sigma$). Using the dot product to measure the similarity between the patterns and spins, the update rule is

$$S^{(t)} = \text{softmax}\left(T \sum_{\sigma \in K}\left(\Xi_\sigma^T \overrightarrow{S_\sigma^{(t-1)}}\right)\right) \Xi. \qquad (7)$$

In practice, however, the dot product has been found to under-perform in modern continuous Hopfield networks compared to Euclidean or Manhattan distances (Millidge et al., 2022). Transformer models in natural language tasks have also seen performance improvements by replacing the dot product with cosine similarity (Henry et al., 2020), again a measure with a more geometric flavour. However, these similarity measures generalise distances between pairs of elements rather than sets of elements.

We therefore use higher-dimensional geometric similarity measures, *cumulative Euclidean distance (ced)* and *Cayley–Menger distance (cmd)*. Let $d_\rho$ be the (Euclidean or Manhattan) distance between pattern $\xi_\rho^\mu$ and spins $S_\rho^{(t)}$ for pattern $\mu$ and spins $\rho \subset \sigma$. Let $K_1^\sigma$ be the subset of $K$ such that all elements in $K_1^\sigma$ are 1–simplex faces of $\sigma$. We define the cumulative Euclidean distance as

$$\text{ced}(\xi_\sigma^\mu, S_\sigma^{(t)}) = \sqrt{\sum_{\rho \in K_1^\sigma} (d_\rho)^2}. \qquad (8)$$

We define $\text{cmd}(\xi_\sigma^\mu, S_\sigma^{(t)})$ as the Cayley–Menger determinant of all $\rho \in K_1^\sigma$, with distances set as $d_\rho$.

**Mixed diluted networks.** A computational concern in the above models is that the number of unique possible $k$–simplices is $\binom{N}{k+1}$, e.g., with $N = 100$ there are approximately $9.89 \times 10^{28}$ possible 50–simplices, compared to just $4,950$ edges (1–simplices) found in a pairwise Hopfield network. If we allow all possible simplices for a simplicial Hopfield network with $N$ neurons, the total number of simplices (excluding 0–simplices, i.e., autapses) will be $\sum_{d=2}^{N} \binom{N}{d}$. Simultaneously, there is also an open question as to how many setwise connections is biologically-realistic to model. We also note that setwise connections can be functionally built from combinations of pairwise connections by introducing additional hidden neurons, as shown by Krotov & Hopfield (2021). Therefore, we might in fact be under-estimating the total number of 'functional' setwise connections, which may appear via common network motifs or 'synapsembles' (Buzsáki, 2010).

Conservatively, we evaluate classes of simplicial Hopfield networks which are low-dimensional, i.e., $\dim(K)$ is small, and where the total number of weighted simplices is not greater than those normally found in a pairwise Hopfield network, i.e., the number of non-zero weights is $\binom{N}{2}$. We randomly choose weights to be non-zero, with each weight of a dimension having an equal probability and according to Table 1. (See Appendix A.4 for a small worked example.) Such random networks have previously been studied in the traditional pairwise case as 'diluted networks' (Treves & Amit, 1988; Bovier & Gayrard, 1993a;b; Löwe & Vermet, 2011). Here we study *mixed diluted networks*, since we use a mixture of connections of different degrees. We believe we are also the first to study such networks beyond pairwise connections, as well as in modern and continuous cases.

**Topology.** Different collections of simplices in a simplicial complex can result in different Euler characteristics (a homotopy invariant property). Table 1 shows this from a parameter perspective via counting only the simplices with non-zero weights. However, even when using the same proportion of 1– and 2–simplices, the choices of which vertices those simplices contain can be different due to randomness. Therefore, the topologies of each network may vary (and so too may their subsequent dynamics and performance). One well-studied and often important topological property in the context of simplicial complexes, homology, counts the number of *holes* in each dimension. In the 0th dimension, this is the number of connected components; in the 1st dimension, this is the number of triangles formed by edges which don't also have a 2–simplex 'filling in' the interior surface of that triangle; in the 2nd dimension, this is the number of tetrahedra formed by triangles which don't also have a 3–simplex 'filling in' the interior volume of that tetrahedron; and so on. The exact number of these holes in dimension $k$ can be calculated by the $k$–*th Betti number*, $\beta_k$ (see Appendix A.5). We calculate these for our networks to observe the relationship between homology and memory capacity.

## 2.3 THEORETICAL MEMORY CAPACITY

**Mixed networks.** Much is already known about the theoretical memory capacity of various Hopfield networks, including those with explicit (Newman, 1988) or implicit (Demircigil et al., 2017) setwise connectivity. However, we wish to point out a somewhat underappreciated relationship between memory capacity and the explicit or implicit number of network connections – which, in the fully-connected network, is determined by the degree of the connections (see Appendix A.6 for proof).

**Corollary 2.2** (Memory capacity is proportional to the number of network connections). *If the connection weights in a Hopfield network are symmetric, then the order of the network's memory capacity is proportional to the number of its connections.*

What happens when there are connections between the same neurons at multiple degrees, i.e., what we call a mixed network? To the best of our knowledge, the theoretical memory capacity of such networks has not been well-studied. However, we found one classical study by Dreyfus et al. (1987) which showed, numerically, adding triplet connections to a pairwise model improved attractivity and memory capacity. Most prior formal studies have only considered connections at single higher degrees (Newman, 1988; Bengtsson, 1990). Although, higher order neural networks have historically considered such mixtures of interactions on different degrees simultaneously (Zhang, 2012), but as regular neural networks (e.g., feed-forward networks), not Hopfield networks. Higher order Boltzmann machines (HOBMs) (Sejnowski, 1986) have also been studied with mixed connections

(Amari et al., 1992; Leisink & Kappen, 1999)[3]. However, HOBMs are unlike Hopfield networks in that they typically have hidden units, are trained differently, and have stochastic neural activations [4]. Modern Hopfield networks also include an implicit mixture of connections of different degrees[5] (but – and see Theorem 1 of Demircigil et al. (2017), which remains unproven – the mixture is unbalanced and not particularly natural, especially for $F(x) = x^n$ when $n$ is odd). Therefore, we include the following result demonstrating fixed points, large basins of attraction (i.e., convergence) to those fixed points in mixed networks, and memory capacity which is linear in the number of fully-connected degrees of connections (a proof is provided in Appendix A.6).

**Lemma 2.3** (Fully-connected mixed Hopfield networks). *A fully-connected mixed Hopfield network based on a $D$–skeleton with $N$ neurons and $P$ patterns has, when $N \to \infty$ and $P$ is finite: (i) fixed point attractors at the memory patterns; and (ii) dynamic convergence towards the fixed point attractors within a finite Hamming distance $\delta$. When $P \to \infty$ with $N \to \infty$, the network has capacity to store up to $(\sum_{d=1}^{D} N^d)/(2 \ln N)$ memory patterns (with small retrieval errors) or $(\sum_{d=1}^{D} N^d)/(4 \ln N)$ (without retrieval errors).*

This naturally comports with Theorem 2 from Demircigil et al. (2017), except here we show an increased capacity in the mixed network, courtesy of Corollary 2.2.

**Mixed diluted networks.** As mentioned earlier, full setwise connectivity is not necessarily tractable nor realistic. Löwe & Vermet (2011) show for pairwise diluted networks constructed as Erdös-Renyi graphs (constructed by including each possible edge on the vertex set with probability $p$) that the memory capacity is proportional to $pN$. Crucial for this result is that the random graph must be asymptotically connected. This makes sense, given that if any vertex was disconnected its dynamics could never be influenced. Empirically, it does seem that a certain threshold of mean connectivity in pairwise random networks is crucial for attractor dynamics (Treves & Amit, 1988).

*Remark* 2.4. By a straightforward generalisation of Löwe & Vermet (2011)'s result, diluted networks constructed as pure Erdös-Renyi hypergraphs may store on the order of $pN^{d-1}$ memory patterns, where $d$ is the degree of the connections.

In the case of an unbounded number of allowable connections, Remark 2.4 would suggest picking as many higher-degree connections as possible when choosing between connections of lower or higher degrees in our mixed diluted networks. However, in the bounded case (our case), we are non-trivially changing the asymptotic behaviour in terms of connectivity and dynamics when we use a mixture of connection degrees. We also need to beware of asymmetries which may arise (Kanter, 1988). This makes the analysis of mixed diluted networks not particularly straightforward (also see Section 4).

## 2.4 NUMERICAL SIMULATIONS AND PERFORMANCE METRICS

Given the large space of possible network settings, in the main text we focus primarily on conditions listed in Table 1. Additional experiments are also shown in Appendix A.8.

Table 1: List of network condition keys (top row), their number of non-zero weights for 1– and 2–simplices (second and third rows), and their 'functional' Euler characteristic ($\chi$, bottom row). $N$ is the number of neurons. $C = (N-1)N$. For simulation, the number of simplices at each dimension are rounded to the nearest integer.

|  | K1 | R$\bar{1}$2 | R$\overline{1}\overline{2}$ | R1$\bar{2}$ | R2 |
|---|---|---|---|---|---|
| 1–simplices | $\binom{N}{2}$ | $0.75\binom{N}{2}$ | $0.50\binom{N}{2}$ | $0.25\binom{N}{2}$ | 0 |
| 2–simplices | 0 | $0.25\binom{N}{2}$ | $0.50\binom{N}{2}$ | $0.75\binom{N}{2}$ | $\binom{N}{2}$ |
| $\chi$ | $N - (1/2)C$ | $N - 0.25C$ | $N$ | $N + 0.25C$ | $N + (1/2)C$ |

---

[3]HOBMs also suffer the same problem as we face here, one of having many high-order parameters between the neurons to keep a track of. Possibly a factoring trick like in Memisevic & Hinton (2010) for HOBMs could be helpful in simplicial Hopfield networks.

[4]Despite this, there are equivalences (Leonelli et al., 2021; Marullo & Agliari, 2021; Smart & Zilman, 2021).

[5]Recall that $\left(\sum_i a_i\right)^b = \sum_i a_i^{b+1}$.

In our numerical simulations, we perform updates synchronously until $E$ is non-decreasing or until a maximum number of steps is reached, whichever comes first. When a simulation concludes we compute the *overlap* (for binary patterns) or *mean squared error (MSE)* (for continuous patterns) of the final spin configuration with respect to all stored patterns using

$$m^\mu = \left| \frac{1}{N} \sum_{i=1}^{N} S_i^{(t)} \xi_i^\mu \right| \qquad\qquad \text{MSE}^\mu = \frac{1}{N} \sum_{i=1}^{N} (S_i^{(t)} - \xi_i^\mu)^2. \qquad (9)$$

We say the network recalls (or attempts to recall) whichever pattern has the largest overlap (where $m^\mu = 1$ indicates perfect recall) or smallest MSE (where $\text{MSE}^\mu = 0$ indicates perfect recall).

## 3 NUMERICAL SIMULATIONS

### 3.1 BINARY MEMORY PATTERNS

After embedding random binary patterns, we started the network in random initial states and recorded the final overlap of the closest pattern. Table 2 shows the final overlaps for traditional simplicial Hopfield networks ($N = 100$). Our simplicial Hopfield networks significantly outperform the pairwise Hopfield networks (K1). In fact, the R1$\overline{2}$ model performs as well at $0.3N$ patterns as the the pairwise network performs on $0.05N$ patterns, a six-fold increase in the number of patterns and more than double the theoretical capacity of the pairwise network, $\sim 0.14N$ (Amit et al., 1985). Surprisingly, Table 3 shows homology accounts for very little of the variance in network performance.

Table 2: Mean $\pm$ standard deviation of overlap distributions ($n = 100$) from traditional simplicial Hopfield networks with varying numbers (top row) of random binary patterns. K1 is the traditional pairwise Hopfield network. R1$\overline{2}$ significantly outperforms K1 at all tested levels (one-way t-tests $p < 10^{-11}$, $F > 50.13$). At all pattern loadings, a one-way ANOVA showed significant variance between the networks ($p < 10^{-20}$, $F > 26.35$). Box and whisker plots shown in Figure 6.

| *No. patterns* | $0.05N$ | $0.1N$ | $0.15N$ | $0.2N$ | $0.3N$ |
|---|---|---|---|---|---|
| K1 | $0.87 \pm 0.18$ | $0.81 \pm 0.16$ | $0.66 \pm 0.10$ | $0.65 \pm 0.10$ | $0.59 \pm 0.08$ |
| R$\overline{1}$2 | $0.96 \pm 0.10$ | $0.94 \pm 0.14$ | $0.82 \pm 0.20$ | $0.71 \pm 0.17$ | $0.64 \pm 0.13$ |
| R$\overline{1}\overline{2}$ | $0.98 \pm 0.10$ | $\mathbf{0.99 \pm 0.03}$ | $0.97 \pm 0.10$ | $0.91 \pm 0.15$ | $0.76 \pm 0.16$ |
| **R1$\overline{2}$** | $\mathbf{1 \pm 0}$ | $\mathbf{0.99 \pm 0.04}$ | $\mathbf{0.99 \pm 0.05}$ | $\mathbf{0.98 \pm 0.08}$ | $\mathbf{0.87 \pm 0.16}$ |
| R2 | $\mathbf{1 \pm 0}$ | $\mathbf{0.99 \pm 0.18}$ | $0.94 \pm 0.18$ | $0.74 \pm 0.29$ | $0.53 \pm 0.23$ |

### 3.2 CONTINUOUS MEMORY PATTERNS

**Energy landscape.** Using Equation 3 and given a set of patterns, a simplicial complex $K$, and an inverse temperature $T^{-1}$, we may calculate the energy of network states. To inspect changes in the energy landscapes of different network conditions, we set $N = 10$ and $P = 10$ random patterns. We performed principle component analysis (PCA) to create a low dimensional projection of the patterns. Then, we generated network states evenly spaced in a $10 \times 10$ grid which spanned the projected memory patterns in the first two dimensions of PCA space. We calculated each state's energy by transforming these points from PCA space back into the $N$–dimensional space, across the network conditions at $T^{-1} = 1, 2, 10$ (Figure 7). At $T^{-1} = 1$, differences between the network conditions' energy landscapes are very subtle. However, at $T^{-1} = 2$ and $T^{-1} = 10$, we see a clear change: those with more 2–simplices possess more sophisticated, pattern-separating landscapes.

**Recall as a function of memory loading.** We tested the performance of our simplicial Hopfield networks by embedding data from the MNIST (LeCun et al., 2010), CIFAR-10 (Krizhevsky & Hinton, 2009), and Tiny ImageNet (Le & Yang, 2015) datasets as memories. We followed the protocol of Millidge et al. (2022) to test recall under increasing memory load as an indication of the networks' memory capacities. To embed the memories, we normalise the pixel values between $0$ and $1$, and treat them as continuous-valued neurons, e.g., for MNIST we have $N = 28 \times 28 = 784$ neurons.

We initialise $S$ as one of the memory patterns corrupted by Gaussian noise with variance $0.5$. After allowing the network to settle in an energy minima, we measured the performance as the fraction of correctly recalled memories (over all tested memories) of the uncorrupted patterns, where 'correct recall' was defined as a sum of the squared difference being $< 50$. In all tests, we used $T^{-1} = 100$. Also see Appendix A.7 for further simulation details.

Figure 2 compares a pairwise architecture, K1, with a higher-order architecture, R1$\overline{2}$. The performance of the K1 networks is comparable to that shown in Millidge et al. (2022), however, R1$\overline{2}$ significantly outperforms K1 across all datasets. Since the MNIST dataset is relatively simple and K1 already performs well, the performance improvement is small, albeit significant. In the CIFAR-10 and Tiny ImageNet datasets, the performance improvements are more noticeable, with most distance functions seeing improvements of $\geq 10\%$ in the fraction of correctly retrieved memories.

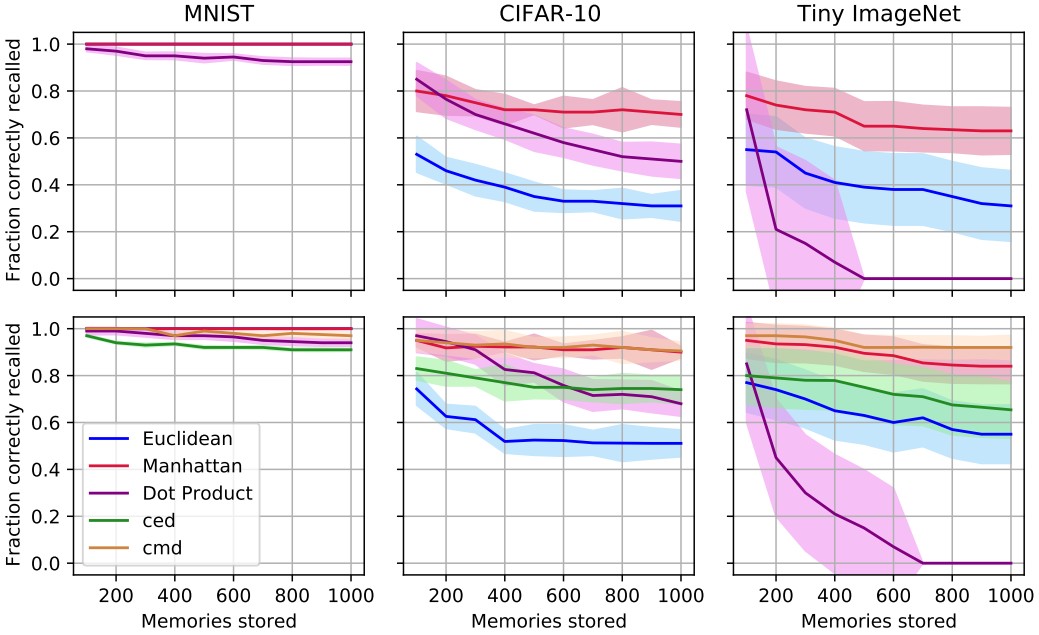

Figure 2: Recall (mean $\pm$ S.D. over 10 trials) as a function of memory loading using the MNIST, CIFAR-10, and Tiny ImageNet datasets, using different distance functions (see legend). Here we compare the performance of modern continuous pairwise networks (top row) and modern continuous simplicial networks (bottom row). The simplicial networks are R1$\overline{2}$ networks (see Table 1 for information). R1$\overline{2}$ significantly outperforms the pairwise network (K1) at all tested levels where there was not already perfect recall (one-way t-tests $p < 10^{-9}$, $F > 16.01$). At all memory loadings, a one-way ANOVA showed significant variance between the networks ($p < 10^{-5}$, $F > 11.95$). Tabulated results are shown in Tables 6, 7, and 8.

Also noticeable in the results for CIFAR-10 and Tiny ImageNet (see Figure 2) is the relatively high performance of the ced and cmd distance measures. Indeed, cmd performs as well or better than the Manhattan distance in our experiments. And both ced and cmd (along with the Euclidean and Manhattan distances) outperform the dot product in CIFAR-10 and Tiny ImageNet at high memory loadings. This further supports the intuition and results of Millidge et al. (2022), that more 'geometric' distances perform better as similarity measures in modern Hopfield networks.

## 4  DISCUSSION

We have introduced a new class of Hopfield networks which generalises and extends traditional, modern, and continuous Hopfield networks. Our main finding is that mixed diluted networks can improve performance in terms of memory recall, even when there is no increase in the number of parameters. This improvement therefore comes from the topology rather than additional information

in the form of parameters. We also show how distance measures of a more 'geometric flavour' can further improve performance in these networks. This simplicial framework (in diluted or undiluted forms) now opens up new avenues for researchers in neuroscience and machine learning. In neuroscience, we can now model how setwise connections, such as those provided by astrocytes and dendrites, may improve memory function and may interact to form important topological structures to guide memory dynamics. In machine learning, such topological structures may now be utilised in improved attention mechanisms or Transformer models, such as in Ramsauer et al. (2021). At the intersection of these fields, we may now further study how the topology of networks in neuroscience and machine learning systems may correspond to one another and share functional characteristics, such as how the activity of 'pairwise' Transformer models have shown similarities to activities in auditory cortex (Millet et al., 2022). Could 'setwise' Transformer models correspond more closely? Or to a more diverse range of cell types? These and related questions are now open for exploration, and may lead to improved performance in applications (Clift et al., 2020).

**Convolution operations and higher-order neural networks.** From the perspective of modern deep learning, considering higher order correlations between downstream inputs to a neuron is quite classical. For example, convolutional neural networks have incorporated specialised setwise operations since their inception (Fukushima, 1980; Lecun et al., 1998), and more general setwise relationships have also been introduced in higher-order neural networks (Pineda, 1987; Reid et al., 1989; Ghosh & Shun, 1992; Zhang, 2012). Although our setwise connections are not explicitly convolutional, they are in one notable sense conceptually similar: they collect information from a particular subset of neurons and only become active when those particular neurons are active in the right way. One of the main differences, however, is that – unlike typical convolution operations – we don't restrict the connection locations to some particular locations or arrangements within the input space. Our results therefore suggest that, in some cases, replacing regular feedforward connections with random convolutions may offer improved performance in some circumstances.

**Improvements and extensions.** Our study focusses on random choices of weighted simplices. What if we choose more carefully? Indeed, it seems quite likely biological setwise connections are not random, and are almost certainly not randomly chosen to replace random pairwise connections.

It now seems natural to study how online weight modulations (e.g., based on spectral theories) could generate new connections between Hopfield networks and, e.g., geometric deep learning. Such modulations may have novel biological interpretations, e.g., spatial and anti-Hebbian memory structures may be modelled by strategically inserting inhibitory interactions (Haga & Fukai, 2019; Burns et al., 2022) between higher simplices (and may also model disinhibition).

**Further analytic studies.** Our numerical results suggest diluted mixed networks have larger memory capacities than pairwise networks. In a fairly intuitive sense, this is not particularly surprising – we are adding degrees of freedom to the energy landscape, within which we may build more stable and nicely-behaved attractors. However, we have not yet proven this increased capacity analytically for the diluted case, only given some theoretical indications as to why this occurs and proven the undiluted case. We hypothesise it is possible to do so using generalised methods from replica-symmetric theory (Amit et al., 1985) or self-consistent signal-to-noise analysis (Shiino & Fukai, 1993), in combination with methods from structural Ramsey theory. The capacity for modern simplicial networks may be on the order of a double-exponential in the number of neurons (since, in the limit of $N \to \infty$, there is an exponential relationship in the number of multispin interactions on top of an exponential relationship in the number of intra-multispin interactions, i.e., both pair-spins and multi-spins can have higher degrees of attraction). This capacity, however, will likely scale nonlinearly with the choice of (random) dilution, e.g., there may be a steep drop in performance around a critical dilution range, likely where some important dynamical guarantees are lost due to an intolerably small number of connections of a particular order.

Even higher orders and diluted mixtures of setwise connections may also be studied. Such networks, per Lemma 2.3, will likely improve their performance as higher-degree connections are added (as shown in Appendix A.8). However, and as implied in Section 2.3, the number and distribution of these connections may need to be careful chosen in highly diluted settings.

REPRODUCIBILITY STATEMENT

To reproduce our results in the main text and appendices, we provide our Python code as supplementary material at `https://github.com/tfburns/simplicial-hopfield-networks`. We have also provided a small worked example in Appendix A.4 to help clarify computational steps in the model construction. Assumptions made in our theoretical results are stated in Section 2.3 and Appendix A.6.

ACKNOWLEDGEMENTS

The first author thanks Milena Menezes Carvalho for graphic design assistance with Figure 1, as well as Robert Tang, Tom George, and members of the Neural Coding and Brain Computing Unit at OIST for helpful discussions. We thank anonymous reviewers for their feedback and suggestions. The second author acknowledges support from KAKENHI grants JP19H04994 and JP18H05213.

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

# A APPENDIX

## A.1 PURPOSE OF STUDYING AND EXTENDING HOPFIELD NETWORKS

In the context of memory more broadly, associative memory is a form of *long-term memory*, memory which can potentially last a lifetime. However, memory also exists on at least two shorter and functionally distinct time-scales: *short-term memory* and *sensory memory*. Short-term or working memory stores information for on the order of tens of seconds, and performs as a limited, passive temporary memory reservoir with which to manipulate and use information (Milner, 1955; Mongillo et al., 2008). Sensory memory operates on even shorter timescales than short-term memory, typically on the order of only seconds. It is where visual (Sperling, 1963; Vogel et al., 2001), auditory (Darwin et al., 1972; Winkler & Cowan, 2005), and other sensory information (Gordon et al., 1993; Lederman & Klatzky, 2009) is first actively 'remembered' – all other information is either immediate sensory information or recalled information. A general schema for forming a long-term memory is therefore to: (1) receive external sensory information; (2) store the features of that sensory information in sensory memory; (3) manage and store one or more features of the sensory information (and/or combine it with other sensory information) in short-term memory; and then (4) consolidate this information into long-term memory. Understanding these processes from a theoretical and computational perspective has several real-world implications, including: (i) it may allow us to create more intelligent machines, by gaining inspiration and insight from biological strategies to store, retrieve, and use long-term associative memories; and (ii) it may help us understand the neurobiological mechanisms (and their implications) for biological memory systems, helping to not only understand related psychological and biological phenomena, but to potentially help identify therapeutic targets for related dysfunction.

Psychological abilities attributed to associative memory in humans and non-human animals are typically said to be any form of long-term memorisation which involves 'pairing' or 'associating' distinct stimuli such that when presented with one stimuli, the subject can recall the other stimuli. Classical examples of this type of associative memory include pairings: name-face pairs (Sperling et al., 2003), object-sound pairs (Preziosi & Coane, 2017), and object-place pairs (Gilbert & Kesner, 2004). These types of associative memories are part of *explicit* or *declarative memory* (Ullman, 2004), i.e., long-term memory that can be explicitly or voluntarily stated or declared. This is in contrast to associative memories which are part of *implicit* or *non-declarative memory*, i.e., long-term memory recalled or used unconsciously or unintentionally. Examples of this type of associative memory are generated by classical conditioning (Maren, 2001; Christian & Thompson, 2003) and operant conditioning (Mackintosh, 1983; McSweeney & Murphy, 2014).

Computational accounts of implicit associative memory have a long and successful history starting from examples like the Rescorla–Wagner model (Rescorla & Wagner, 1972). Today, this research has grown into the computational field of reinforcement learning (Daw & Doya, 2006). In comparison to implicit associative memory, explicit associative memory seems more computationally sophisticated, as suggested by its complex biological bases (Chaudhuri & Fiete, 2016; Clopath et al., 2017; Mau et al., 2020).

Soon after the proposal of Marr (1971), one of the first and most influential computational models of (explicit) associative memory was the Hopfield model (Hopfield, 1982), which we study and extend here. A nice feature of this model is that in the basic case of embedding a single memory, it is easy to see there exists a choice of threshold for every neuron whereby any partial activation of the memory will lead to activation of all its other members. This therefore generates an attractor dynamic around the fixed point of the stored memory, which is comparable to the neuronal assembly attractor dynamics seen in the hippocampus (Wills et al., 2005; Pfeiffer & Foster, 2015; Rebola et al., 2017).

Hippocampus is probably the foremost brain structure involved in memory. It, together with the surrounding entorhinal, perirhinal, and parahippocampal cortices, is especially important for explicit memory (Scoville & Milner, 1957; Milner, 1966; Squire, 1992). It is a necessary structure for the initial formation and learning of explicit memory, acting as a short-term memory for later long-term consolidation, thought to occur in cortex (Squire et al., 1989; Sutherland & Rudy, 1989). Classically, we think these capacities are mainly achieved via Hebbian learning and long-term potentiation (Bliss & Gardner-Medwin, 1973; Gustafsson & Wigström, 1988). However, there is now an increasing literature which shows how other mechanisms may help to achieve these memory functions (see Appendix A.2 for examples).

As a complete computational account of long-term memory storage, the classical Hopfield model encounters challenges. As discussed in Section 1 of the main text, the memory capacity of the classical Hopfield network is linear in the number of neurons $N$, specifically: approximately $0.14N$ patterns may be stored before spurious attractors overwhelm workable levels of memory recall (Amit et al., 1985; McEliece et al., 1987; Bruck & Roychowdhury, 1990), and this capacity diminishes further when the patterns are statistically or spatially correlated (Löwe, 1998), in sparse connectivity regimes (Treves & Amit, 1988; Löwe & Vermet, 2011), and in combination (Burns et al., 2022). Biological networks typically have very sparse connectivity (Minai & Levy, 1993; Lansner, 2009; Barth & Poulet, 2012) and everyday memory items typical share many statistical features, may have sophisticated inter-relations, and are spatially or semantically correlated in non-trivial structures (Constantinescu et al., 2016; Aronov et al., 2017; Bellmund et al., 2018; Bao et al., 2019; Park et al., 2021; Griesbauer et al., 2022). Despite this, humans can remember very high-fidelity information of thousands of statistically similar images (Standing, 1973; Brady et al., 2008) and human faces (Jenkins et al., 2018), tens of thousands of linguistic items (Brysbaert et al., 2016), and more than 100,000 digits of the number $\pi$ (Bellos, 2015) – all, seemingly, without dramatically sacrificing or over-writing other memories.

Although modern Hopfield networks have substantially increased theoretical memory capacity (Krotov & Hopfield, 2016; Demircigil et al., 2017), the combined biological and psychological evidence mentioned above, along with the finite (if large) number of brain cells (Herculano-Houzel, 2009) and energetic demands of maintaining them and their inter-connections (Bordone et al., 2019), suggest there may be more to the neurophysiological and computational story. Furthermore, even if we find that the theoretical memory capacity should still be high according to a Hopfield interpretation, capacity can be considered as a measure of undesired interferences between memories, and thus may be maximised for cognitive convenience or speed and accuracy of memory recall.

Nevertheless, problems and criticisms don't detract from the usefulness and importance of the Hopfield model or its modern variations in the study of memory systems (Sathasivam & Wan Abdullah, 2008; Rizzuto & Kahana, 2001; Weber et al., 2017), usefulness in machine learning applications (Widrich et al., 2020; Seidl et al., 2022), contribution to more general machine learning (Sharma et al., 2022; Hoover et al., 2022), or connection between biology and machine learning (Chaudhuri & Fiete, 2019; Tyulmankov et al., 2021; Kozachkov et al., 2022). There are even more opportunities to build upon this substantial foundation to create more sophisticated computational models of associative memory. Notably, much work has improved the efficiency and capacity of the Hopfield network (Storkey, 1997; Hopfield, 2008; Krotov & Hopfield, 2016; Gripon & Berrou, 2011; Mofrad & Parker, 2017). Other work has focused on achieving sparse representations (Kim et al., 2017; Hoffmann, 2019) or including other forms of biological realism (Watson et al., 2011a;b; Woodward et al., 2015; Burns et al., 2022). The current work contributes to developments in sparsity, biological realism, and memory capacity.

## A.2 Setwise connections and modulations are bountiful in biology

Setwise connections are not limited to the case, as one might expect, of multiple synaptic contacts between pairs or sets of cells (Jones & Powell, 1969; Sorra & Harris, 1993; Geinisman et al., 2001; Lee et al., 2013; Rigby et al., 2022), which may result in multiplicative interactions (Poleg-Polsky & Diamond, 2016; Reuveni et al., 2017; Groschner et al., 2022) or form of functional synaptic clusters (Kavalali et al., 1999; Bloss et al., 2018; Pulikkottil et al., 2021; Hedrick et al., 2022). Other examples (some of which are illustrated in Figure 3) include the wide spatial dispersion of certain neurotransmitters (Rusakov & Kullmann, 1998; Arbuthnott & Wickens, 2007; Kato et al., 2022), dendro-dendritic synapses (Pinault et al., 1997; Didier et al., 2001; Brombas et al., 2017), persistently-connected neuronal assembly structures or 'synapsembles' (Buzsáki, 2010; Papadimitriou et al., 2020), distributed persistent activity during activities such as motor planning and working memory (Guo et al., 2017; Hart & Huk, 2020), neuroglial modulations of neurotransmitter release probabilities across multiple neurons or synapses (Min et al., 2012; Covelo & Araque, 2018; Chipman et al., 2021), 'tripartite' astrocyte–neuron synapses (Araque et al., 1999; Perea et al., 2009), astrocytic coding (Doron et al., 2022), and during dendritic integration at the level of individual neurons (Golding et al., 2002; Etherington et al., 2010). Modulations of and interactions between such connections are illustrated in Figure 4.

It is also possible to 'functionally' construct setwise connections through only pairwise synapses, as shown in Krotov & Hopfield (2021). In one sense, this kind of pairwise-based reconstruction of setwise connections could also be thought of as similar to results from Poirazi et al. (2003), who showed how multi-layer artificial neural networks can approximate more biologically-sophisticated model neurons with dendrites. Indeed, many of the known and suggested computational features of dendritic integration (Poirazi & Papoutsi, 2020; Chavlis & Poirazi, 2021) may be considered as highly specialised and sophisticated forms of convolutions or setwise interactions.

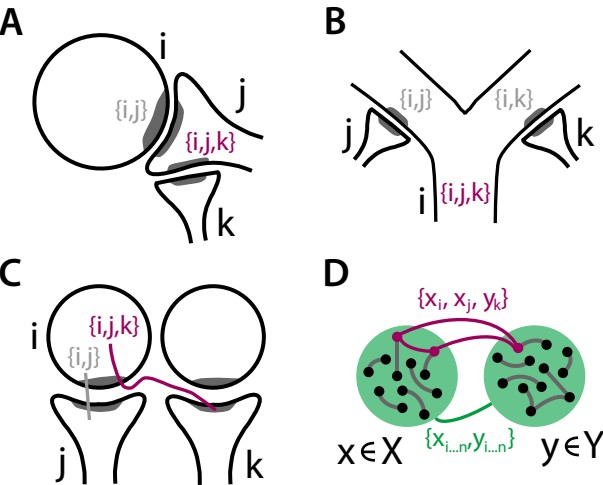

Figure 3: **A.** Multi-synaptic bouton. **B.** Dendritic integration. **C.** Extra-synaptic neurotransmitter diffusion. **D.** Functional connections between neural assemblies.

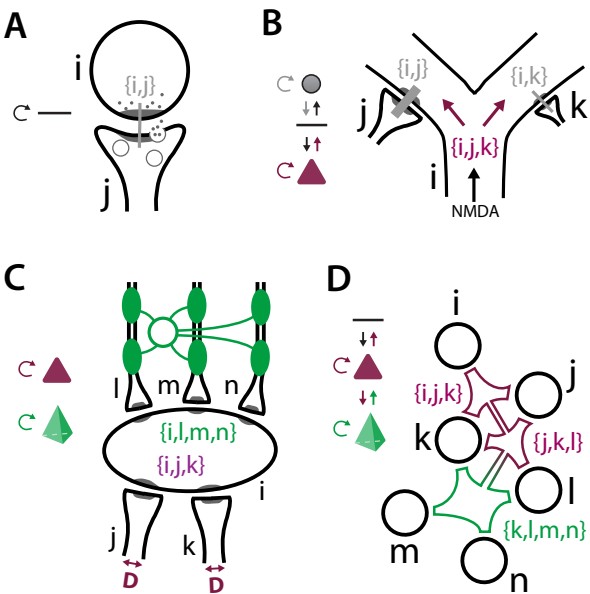

Figure 4: **A.** Neurotransmitter depletion at a synapse. **B.** NMDA-spikes and unequal synaptic strengths in dendritic integration. **C.** Transmission speed plasticity using myelination (top) and axon diameter (bottom) to affect 'temporal'/functional setwise influence in a post-synaptic cell. **D.** Astrocytic messaging, between both neurons and astrocytes.

### A.3 OPTIONS FOR MODELLING SETWISE CONNECTIVITY IN NEURAL NETWORKS AND WHY WE CHOOSE SIMPLICIAL COMPLEXES

In geometric and topological artificial intelligence and machine learning, recent advances have been realised by utilising higher dimensional analogues of graphs such as simplicial complexes (Ebli et al., 2020; Roddenberry et al., 2021), cube complexes (Burns & Tang, 2022), cell complexes (Hajij et al., 2020; Bodnar et al., 2021), and hypergraphs (Feng et al., 2019; Xu et al., 2022). Unlike graphs, these structures can naturally represent higher-degree, setwise relationships. However, not all structures are appropriate for all systems (Spivak, 2009; Rosas et al., 2022).

Why do we choose to model our collections of setwise connections as weighted simplicial complexes and not use general cell complexes or hypergraphs? There are three main reasons:

1. *Simplicial complexes allow all possible setwise relations to exist.* Simplices, by construction, may span any number of vertices. This means any possible combination of neurons may share a common, setwise weight. This is also possible in hypergraphs, but not possible in all complexes, e.g., cube complexes may not include triangles. The 'natural' complex for modelling all possible setwise relationships is therefore a simplicial one. Since we also wish for our setwise weights to be symmetrical (i.e., have the same value when updating the spin with respect to each constituent neuron), it is unnecessary to include any more than one unique object per setwise relationship. This also makes the choice of a simplex suitable, since there can only be one unique simplex for a given set of vertices – which is also the case for edges in undirected hypergraphs (but we find these are less suitable).

2. *The sub-edge problem of hypergraphs makes them less suitable.* Hypergraphs are graphs where edges may contain any number of unique vertices from the vertex set. In a sense, these are a more general structure than simplicial complexes and lack downward closure, e.g., if the edge $\{1, 2, 3\}$ exists, edges such as $\{2, 3\}$ or $\{1\}$ do not necessarily exist, whereas if $\{1, 2, 3\}$ was a simplex in a simplicial complex, simplices $\{2, 3\}$ and $\{1\}$ exist. However, hypergraphs do not have well-defined 'sub-edges' (described as the 'sub-edge problem' in Remark 3.5 of Spivak (2009)). This has the consequence of defining interactions between 'levels' of hyperedges (setwise relationships) in hypergraphs slightly awkward. In contrast, simplicial complexes have a well-defined hierarchy of setwise relationships, partly due to the downward closure condition.

3. *Downward closure of setwise connections is biologically plausible.* Another benefit of downward closure in simplicial complexes is that it currently seems better supported from the perspective of biological plausibility (also see Appendix A.2). For example, although it can happen that a setwise connection (anatomical or functional) between neurons could exist without any underlying pairwise connections, the typical machinery used to create such setwise connections is sufficiently local to assume that, because of the functional local modularity of connections in the brain (Kaiser & Hilgetag, 2006; Chen et al., 2013; Müller et al., 2020; Ercsey-Ravasz et al., 2013), there is a high probability of these neurons having a pairwise connection simply due to proximity.

As an additional practical benefit, simplicial complexes are currently more well-studied than structures such as hypergraphs (at least in some areas, e.g., spectral theories or (co)homology, which are of natural interest here), meaning that we can also take advantage of the relative maturity of the field in those areas – admittedly, we use very few advanced methods or properties in an essential way in this study, although we hope to do so in future studies, having now introduced an initial interpretation of simplicial Hopfield networks and begun exploring some of their potential benefits. However, it will also be interesting to see what differences can be found between hypergraphic and simplicial Hopfield networks, and perhaps which provides a closer approximation to biology or which shows improved performance on certain tasks.

Among other possibilities, the weight of a simplex $w(\sigma)$ could be modulated by the local energy of its spins $S_\sigma$, its coface's spins, or of those of simplices in the same dimension as $\sigma$ which are 'neighbouring' (in the Hodge Laplacian sense (Lim, 2020; Schaub et al., 2020)). These interactions could take many different mathematical forms (Petri & Barrat, 2018; Ebli et al., 2020; Roddenberry et al., 2021; Rosas et al., 2022; Santoro et al., 2022). Neurobiologically, these interactions could represent neural–glia interactions, glia–glia interactions, nonlinear dendritic integration (especially dendritic spikes and shunting), neurotransmitter–neuromodulator interactions, or hierarchical as-

sembly operations and dynamics, to name a few (see Appendix A.2 for illustrations and further information).

The downward closure of simplicial complexes could be seen as a disadvantage. For example, when including simplices of high dimension, we are also forced to limit our choices of simplices if we wish to maintain the simplicial structure. Again, whenever a $k$–simplex exists in $K$, all its faces must also exist, e.g., if a triangle exists, so must its surrounding edges and their surrounding vertices. If any constituent simplex is missing, the structure of the simplicial complex is broken and in our case would become an undirected, weighted hypergraph. Instead, in our simulations, we prefer to interpret 'missing simplices' as merely functionally 'missing-in-action' by setting their weights to zero if we do not wish to include them in the model. This has the consequence of having no mathematical effect on our update rules while retaining the convenience of a simplicial structure.

### A.4    A SMALL WORKED EXAMPLE

Consider a small example of just $P = 3$ memory patterns embedded in a simplicial Hopfield network on $N = 6$ neurons. First, let's consider a 3–skeleton, i.e., a network without any 'dilution' or 'missing weights' up to $D = 3$. Such a network will have functional connections totalling

$$\sum_{d=2}^{D+1} \binom{N}{d} = \binom{6}{2} + \binom{6}{3} + \binom{6}{4} = 15 + 20 + 15 = 50. \tag{10}$$

We typically do not include functional self-connections (autapses) – although Hopfield networks with such networks have been studied (Folli et al., 2017; Rocchi et al., 2017; Gosti et al., 2019). In simplicial Hopfield networks, such self-connections correspond to 0–simplices, i.e., vertices. While these vertices do exist in the underlying simplicial complex $K$, we set their associated weights to 0.

Given $N = 6$, the 3–skeleton in this example is

$$
\begin{aligned}
K_0 =& \{\{1\}, \{2\}, \{3\}, \{4\}, \{5\}, \{6\}\}, \\
K_1 =& \{\{1,2\}, \{1,3\}, \{1,4\}, \{1,5\}, \{1,6\}, \{2,3\}, \{2,4\}, \{2,5\}, \{2,6\}, \{3,4\}, \{3,5\}, \\
& \{3,6\}, \{4,5\}, \{4,6\}, \{5,6\}\}, \\
K_2 =& \{\{1,2,3\}, \{1,2,4\}, \{1,2,5\}, \{1,2,6\}, \{1,3,4\}, \{1,3,5\}, \{1,3,6\}, \{1,4,5\}, \{1,4,6\}, \\
& \{1,5,6\}, \{2,3,4\}, \{2,3,5\}, \{2,3,6\}, \{2,4,5\}, \{2,4,6\}, \{2,5,6\}, \{3,4,5\}, \{3,4,6\}, \\
& \{3,5,6\}, \{4,5,6\}\}, \\
K_3 =& \{\{1,2,3,4\}, \{1,2,3,5\}, \{1,2,3,6\}, \{1,2,4,5\}, \{1,2,4,6\}, \{1,2,5,6\}, \{1,3,4,5\}, \\
& \{1,3,4,6\}, \{1,3,5,6\}, \{1,4,5,6\}, \{2,3,4,5\}, \{2,3,4,6\}, \{2,3,5,6\}, \{2,4,5,6\}, \\
& \{3,4,5,6\}\}, \\
K =& \emptyset \cup K_0 \cup K_1 \cup K_2 \cup K_3.
\end{aligned}
\tag{11}
$$

Set three patterns as

$$
\begin{aligned}
\xi^1 =& (-1, +1, -1, +1, -1, +1), \\
\xi^2 =& (+1, -1, +1, -1, -1, +1), \\
\xi^3 =& (-1, -1, -1, +1, +1, +1).
\end{aligned}
\tag{12}
$$

For all $\sigma \in K_0$, we set $w(\sigma) = 0$. For all higher dimensions, we set

$$w(\sigma) = \frac{1}{N} \sum_{\mu=1}^{P} \xi_\sigma^\mu. \tag{13}$$

For example,

$$
\begin{aligned}
w(\{1,3\}) =& 1/6 \cdot ((-1 \cdot +1) + (+1 \cdot +1) + (-1 \cdot -1)) = 1/6, \\
w(\{3,5,6\}) =& 1/6 \cdot ((-1 \cdot -1 \cdot +1) + (+1 \cdot -1 \cdot +1) + (-1 \cdot +1 \cdot +1)) = -1/6, \\
w(\{2,4,5,6\}) =& 1/6 \cdot ((+1 \cdot +1 \cdot -1 \cdot +1) + (-1 \cdot -1 \cdot -1 \cdot +1) + (-1 \cdot +1 \cdot +1 \cdot +1)) \\
=& -1/2.
\end{aligned}
\tag{14}
$$

Given a set of spins $S^{(t)}$ at a time-step $t$, the network will evolve according to Equation 2, minimising the energy shown in Equation 1. The energy function consists of a sum of products of the weights with the product of their respective spins, e.g., if $S^{(t)} = (+1, +1, -1, +1, -1, -1)$,

$$E = -\left( \cdots + w(\{1,3\}) \cdot S^{(t)}_{\{1,3\}} + \cdots + w(\{3,5,6\}) \cdot S^{(t)}_{\{3,5,6\}} + \cdots + w(\{2,4,5,6\}) \cdot S^{(t)}_{\{2,4,5,6\}} + \cdots \right)$$
$$= -\left( \cdots + 1/6 \cdot -1 + \ldots -1/6 \cdot -1 + \ldots +1/2 \cdot 1 + \ldots \right).$$

(15)

In this 3–skeleton case, then, the network's energy function can also be written as

$$E = -\left[ \frac{1}{2} \sum_{i,j} \left( \frac{1}{N} \sum_{\mu=1}^{P} \xi_i^\mu \xi_j^\mu \right) S_i^{(t)} S_j^{(t)} \right.$$
$$+ \frac{1}{3} \sum_{i,j,k} \left( \frac{1}{N} \sum_{\mu=1}^{P} \xi_i^\mu \xi_j^\mu \xi_k^\mu \right) S_i^{(t)} S_j^{(t)} S_k^{(t)}$$
$$\left. + \frac{1}{4} \sum_{i,j,k,l} \left( \frac{1}{N} \sum_{\mu=1}^{P} \xi_i^\mu \xi_j^\mu \xi_k^\mu \xi_l^\mu \right) S_i^{(t)} S_j^{(t)} S_k^{(t)} S_l^{(t)} \right].$$

(16)

Essentially, the energy function is similar to a sum of the energy functions of Krotov & Hopfield (2016) with all possible $k$–neuron connections, but where the weights of those connections are independent of each other for each level of interaction, making each connection and each level more controllable. The memory capacity of this type of simiplicial Hopfield network is discussed in Section 2.3 and Appendix A.6. However, one of our main contributions is the findings related to the 'diluted' case, i.e., where more than just the 0–simplices have their weights set to 0. Indeed, these are the cases we mainly evaluate in Section 2.4.

Following on with the same example as above, we can create a diluted simplicial Hopfield network based on $K$. For example, if we chose to limit ourselves to $\binom{6}{2} = 15$ parameters, we could choose to apportion one third of these parameters to each dimension, i.e., set weights only for a subset $K' \subset K$, e.g.,

$$K'_1 = \{\{1,2\}, \{1,6\}, \{2,3\}, \{2,4\}, \{5,6\}\},$$
$$K'_2 = \{\{1,2,3\}, \{1,2,6\}, \{1,3,4\}, \{3,4,5\}, \{3,4,6\}\},$$
$$K'_3 = \{\{1,3,5,6\}, \{1,4,5,6\}, \{2,3,4,5\}, \{2,3,4,6\}, \{2,3,5,6\}\},$$
$$K' = K'_1 \cup K'_2 \cup K'_3.$$

(17)

In our numerical simulations, the choice of which connections to keep is entirely random. By analogy, we can think of this dilution procedure as a naïve solution to the following (fairly contrived) communications problem: Imagine we are tasked with increasing the speed at which a deliberative body of people, e.g., a very large committee, comes to its decisions. Currently, each committee member has individual channels of communication with every other member. This is good for high-fidelity, accurate, and nuanced conversations between members, but not so good for efficiency or speed of decision-making. For example, if a certain block of members consistently vote similarly, it would perhaps be quicker for those members to communicate as a group to check what their majority opinion is rather than all members individually communicating with every other member one at a time. Conversely, when two members consistently vote differently or are active members within distinct voting blocks (and especially if their votes are often tie-breakers), perhaps those two members ought to regularly discuss matters privately and in detail. Our naïve solution is to randomly replace some individual channels of communication with small group communication channels. Possibly by performing a survey of members or observing patterns in their voting or communications, we could come up with a better strategy. However, in this analogy, it appears the naïve solution works reasonably well (see results in Section 3 of the main text). We think a deserving next step will be determine better strategies, perhaps based on or accounting for the overall memory structure and correlations between memory items.

A.5 SIMPLICIAL HOMOLOGY

Simplicial homology allows us to precisely count the number of 'holes' in each dimension of a simplicial complex by calculating the $k$–dimensional Betti number, $\beta_k$. A related topological property which is particularly useful when studying low-dimensional objects (e.g., classification of surfaces) is the Euler characteristic, which for a simplicial complex can be calculated by $\chi(K) = \sum_{i=0}^{\infty} (-1)^k |K_k|$, i.e., it is an alternating sum which 'balances' out the number of holes in odd and even dimensions. It is related to the Betti numbers insofar as the Euler characteristic is also given by $\chi(K) = \sum_{i=0}^{\infty} (-1)^k \beta_k$. Note, however, that $|K_k| \neq \beta_k$. As such, although the Euler characteristic can be used for comparing the topologies of two simplicial complexes, it is not as informative (with respect to holes) as homology (although the latter is more costly to compute, as we will now see).

**$k$-chains and boundaries.** The group of $k$-chains is a free Abelian group with the basis of $K_k$,

$$C_k = C_k(K) := \mathbb{Z}K_k := \left\{ \sum_{\sigma \in K_k} \alpha_\sigma \sigma \mid \alpha_\sigma \in \mathbb{Z} \right\}.$$

The boundary (difference between the 'end points') of a face $\sigma$ in dimension $k$ is

$$\partial_k(\sigma) := \sum_{j \in \sigma} \operatorname{sign}(j, \sigma)(\sigma \setminus j).$$

where $\operatorname{sign}(j, \sigma) = (-1)^{i-1}$, where $j$ is the $i$-th element of $\sigma$ (ordered) and $\sigma \setminus j := \sigma \setminus \{j\}$.

**Example 1.** Consider the simplicial complex

$$K = \{\{1, 2, 3\}, \{1, 2\}, \{1, 3\}, \{2, 3\}, \{1\}, \{2\}, \{3\}, \emptyset\}. \tag{18}$$

For $k = 2$, we have $K_2 = \{\{1, 2, 3\}\}$ and $C_2 = \{\{1 \cdot \{1, 2, 3\}\}\}$. Let us calculate the boundary of $\sigma = \{1, 2, 3\}$. First, we calculate the respective sign functions:

$$\operatorname{sign}(1, \sigma) = (-1)^{i-1} = (-1)^{1-1} = (-1)^0 = 1$$
$$\operatorname{sign}(2, \sigma) = (-1)^{i-1} = (-1)^{2-1} = (-1)^1 = -1$$
$$\operatorname{sign}(3, \sigma) = (-1)^{i-1} = (-1)^{3-1} = (-1)^2 = 1.$$

Using these values, we may calculate the boundary by

$$\begin{aligned}
\partial_2(\sigma) &= \operatorname{sign}(1, \sigma)(\sigma \setminus 1) & &+ \operatorname{sign}(2, \sigma)(\sigma \setminus 2) & &+ \operatorname{sign}(3, \sigma)(\sigma \setminus 3) \\
&= (1)(\sigma \setminus 1) & &+ (-1)(\sigma \setminus 2) & &+ (1)(\sigma \setminus 3) \\
&= (1)(\{2, 3\}) & &+ (-1)(\{1, 3\}) & &+ (1)(\{1, 2\}) \\
&= \{2, 3\} & &- \{1, 3\} & &+ \{1, 2\}.
\end{aligned}$$

The boundary of $\{1, 2, 3\}$ is $\{2, 3\} - \{1, 3\} + \{1, 2\}$. Notice, this is a cycle.

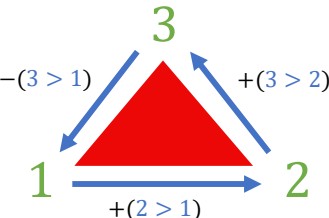

Figure 5: Geometric realisation of the simplicial complex in equation 18 and the boundary of $\{1, 2, 3\}$.

**Chain complex.** The $k$-th boundary mapping, $\partial_k$, is the map $C_k(K) \to C_{k-1}(K)$. If $k > m-1$ or $k < -1$, then $C_k(K) := 0$ and $\partial_k := 0$, where $m$ is the number of 0–simplices. Based on this, the chain complex of $K$ is

$$0 \to C_{n-1}(K) \xrightarrow{\partial_{n-1}} \ldots \xrightarrow{\partial_2} C_1(K) \xrightarrow{\partial_1} C_0(K) \xrightarrow{\partial_0} C_{-1}(K) \to 0.$$

We define $\partial^2 := \partial \circ \partial = 0$. For example, $\partial_{i-1} \circ \partial_i = 0$. This has the consequence of making the boundary of, say, a solid tetrahedron (3–simplex) a set of oriented triangles (2–simplices) with a 'net flow' of 0 (similar to Stokes' curl theorem in calculus).

**Example 2.** Consider the following simplicial complex and its chain complex:

$$K = \{\{1,2\}, \{1\}, \{2\}, \{3\}, \{4\}, \emptyset\},$$

$$0 \to C_1(K) \xrightarrow{\partial_1} C_0(K) \xrightarrow{\partial_0} C_{-1}(K) = 0.$$

The boundary map $\partial_1$ is $\{1,2\} \mapsto \{2\} - \{1\}$ and all faces in $K_0$ are mapped to the empty set by $\partial_0$.

**Homology.** We can now see that our $k$-cycles are $Z_k = \ker \partial_k$ ($k$-chains $\alpha$ where $\partial_k(\alpha) = 0$). Whereas, the $k$-boundaries are $B_k = \operatorname{im} \partial_{k+1}$ ($k$-chains in the image of $\partial_{k+1}$). Notice, $B_k \subset Z_k$.

The (reduced) $k$-homology of $K$ is the Abelian group

$$\widetilde{H_k}(K) := Z_k / B_k,$$

and we define $\widetilde{H_{m-1}}(K) := \ker \partial_{m-1}$ for $k > m-1$ and $\widetilde{H}_k(K) := 0$ for $k < 0$.

The $k$-th Betti number (number of topological holes) is

$$\begin{aligned}
\beta_k &= \dim\left(\widetilde{H_k}\right) \\
&= \dim\left(Z_k\right) - \dim\left(B_k\right) \\
&= \operatorname{nullity}\left(\partial_k\right) - \operatorname{rank}\left(\partial_{k+1}\right).
\end{aligned}$$

Note, nullity $\partial_0 = \dim(C_0)$.

**Example 3.** Consider the simplicial complex

$$K = \{\{1,2\}, \{1,3\}, \{2,3\}, \{1\}, \{2\}, \{3\}, \emptyset\},$$

which is the same as that depicted in Figure 5 but without the filled-in triangle. The chain complex is

$$0 \to \mathbb{Z}^3 \xrightarrow[\partial_1]{\begin{array}{c} \phantom{\{1\}} \{1,2\}\ \{1,3\}\ \{2,3\} \\ \{1\}\quad -1\quad\ -1\quad\ \ 0 \\ \{2\}\quad\ \ 1\quad\ \ \ 0\quad -1 \\ \{3\}\quad\ \ 0\quad\ \ \ 1\quad\ \ \ 1 \end{array}} \mathbb{Z}^3 \xrightarrow[\partial_0]{(0\ \mathrm{map})} 0.$$

We may compute its Betti numbers by

$$\begin{aligned}
\beta_0 &= \operatorname{nullity}\left(\partial_0\right) - \operatorname{rank}\left(\partial_{0+1}\right) \\
&= 3 - 2 = 1 \\
\beta_1 &= \operatorname{nullity}\left(\partial_1\right) - \operatorname{rank}\left(\partial_{1+1}\right) \\
&= 1 - 0 = 1,
\end{aligned}$$

and, by definition, $\beta_{>1} = 0$ in this example.

## A.6  PROOFS

The following are our proofs for statements in the main text.

**Corollary A.1** (Memory capacity is proportional to the number of network connections). *If the connection weights in a Hopfield network are symmetric, then the order of the network's memory capacity is proportional to the number of its connections.*

*Proof.* Let $d$ be the degree of connections in a Hopfield network with $N$ neurons. The explicit or implicit number of connections in such a network is $N^d$. By a simple counting argument, the number of repeated connections between any set of $d$ neurons is interpreted as $d!$. By Newman (1988); Demircigil et al. (2017), the order of such a network's memory capacity is $N^{d-1}$. So the following relationship holds:

$$\frac{N^{d-1}}{N^d/d!} \cdot \frac{N}{d!} = \frac{d!}{N} \cdot \frac{N}{d!} = 1.$$

$\square$

**Lemma A.2** (Fully-connected mixed Hopfield networks). *A fully-connected mixed Hopfield network based on a $D$–skeleton with $N$ neurons and $P$ patterns has, when $N \to \infty$ and $P$ is finite: fixed point attractors at the memory patterns and dynamic convergence towards the fixed point attractors within a finite Hamming distance $\delta$. When $P \to \infty$ with $N \to \infty$, the network has capacity to store up to $(\sum_{d=1}^D N^d)/(2 \ln N)$ memory patterns (with small retrieval errors) or $(\sum_{d=1}^D N^d)/(4 \ln N)$ (without retrieval errors).*

*Proof.* Let $K$ be a $D$–skeleton. Let $S_i^{(t)}$ be the spin of neuron $i$ at time-step $t$, and let the spin correspond to a stored pattern, $S_i^{(t)} = \xi_i^1$. To be in a fixed point, the local field $h_i$ applied to $i$ must satisfy the inequality $S_i^{(t)} h_i > 0$, meaning the local field being applied to the neuron must be of the same sign as the present spin.

In the case of the mixed network, based on the simplicial Hopfield Equations 1–2, the local field is

$$h_i\{S^{(t)}\} = \frac{1}{|K_1|} \sum_{\sigma \in K_1} \sum_{\mu=1}^{P} \xi_i^\mu \xi_{\sigma\backslash i}^\mu S_{\sigma\backslash i}^{(t-1)} + \frac{1}{|K_2|} \sum_{\sigma \in K_2} \sum_{\mu=1}^{P} \xi_i^\mu \xi_{\sigma\backslash i}^\mu S_{\sigma\backslash i}^{(t-1)} + \cdots$$
$$= \sum_{d=1}^{D} \left( \frac{1}{|K_d|} \sum_{\sigma \in K_d} \sum_{\mu=1}^{P} \xi_i^\mu \xi_{\sigma\backslash i}^\mu S_{\sigma\backslash i}^{(t-1)} \right). \tag{19}$$

Because $K$ is a $D$–skeleton, each dimension $K_d$ will have $\binom{N}{d}$ elements. Using Stirling's Approximation for the binomial coefficient, $\binom{N}{d} \approx N^d/d!$ (for $N \gg d$, which holds if $\dim(K)$ is small, which we argue it should normally be for both computational and biological reasons), we can simplify Equation 19 slightly

$$h_i\{S^{(t)}\} = \sum_{d=1}^{D} \left( \frac{d!}{N^d} \sum_{\sigma \in K_d} \sum_{\mu=1}^{P} \xi_i^\mu \xi_{\sigma\backslash i}^\mu S_{\sigma\backslash i}^{(t-1)} \right). \tag{20}$$

To analyse the stability of a pattern, we set $S_i^{(t)} = \xi_i^1$, where the choice of 1 is arbitrary (since the weights are symmetric). Substituting $\xi^1$ for $S$ and Equation 20 for $h$, the inequality we must satisfy for $i = 1$ becomes

$$\xi_1^1 h_1 = \sum_{d=1}^{D} \left( \frac{d! \prod_m^d (N-m)}{N^d} + \frac{d!}{N^d} \sum_{\sigma \in K_d} \sum_{\mu=2}^{P} \xi_1^1 \xi_1^\mu \xi_{\sigma\backslash i}^\mu \xi_{\sigma\backslash i}^1 \right) > 0. \tag{21}$$

Notice in Equation 21 we decomposed the summation over patterns into *signal* terms (for the pattern we are analysing) and *noise* terms (for the contribution of all other patterns). In the limit of $N \to \infty$, the signal terms are fixed numbers (of order 1) and, by the Central Limit Theorem, since the noise terms are sums of random numbers (essentially, random walks), they will have means of 0 and standard deviations of

$$\frac{d!}{N^d} \sqrt{(P-1) \prod_m^d (N-m)}, \tag{22}$$

which we can approximate as $\sqrt{P/N^d}$. We can see that as $d$ increases, the noise terms reduce in variance. However, if $P$ remains fixed and $N$ is sufficiently large, the noise terms become negligible compared to the signal terms. This therefore guarantees that every pattern will be a *fixed point*.

Furthermore, these fixed points will remain highly stable against random noise. Suppose we randomly flip a finite number $\delta$ of spins away from a pattern $\xi = S$ (a fixed point). The signal terms' strengths are reduced by $2\delta$ but still of order 1, whereas the noise terms remain of order $N^{-1/2}$. Therefore, states within Hamming distance $\delta$ away from $\xi$ will *converge* to the fixed point.

Now let $P \to \infty$ with $N$. The total variance of the noise terms is

$$v = \sum_{d=1}^{D} \sqrt{P/N^d}. \tag{23}$$

The probability of stability of neuron $i$ in Equation 21 is the probability that the noise terms are larger than $-1$; at $\leq -1$ they will overcome the signal terms. This probability is

$$\Pr(\xi_1^1 h_1 > 0) = \frac{1}{\sqrt{2\pi v^2}} \int_{-1}^{\infty} dx \, \exp\left(-\frac{x^2}{2v^2}\right) = \frac{1}{2}\left[1 + \text{erf}\left(\sqrt{\frac{1}{2v^2}}\right)\right], \tag{24}$$

where

$$\text{erf}(x) = \frac{2}{\sqrt{\pi}} \int_0^x dt e^{-t^2}. \tag{25}$$

For small $v$ (which this is, especially as $d$ increases and relative to the signal), the value of the error function in Equation 24 will be large and can therefore be approximated (Gradshteyn & Ryzhik, 2007) as

$$\text{erf}(x) \approx 1 - \frac{1}{\sqrt{\pi}x} e^{-x^2}. \tag{26}$$

We can now approximate Equation 24 as

$$\Pr(\xi_1^1 h_1 > 0) \approx 1 - \sqrt{\frac{z}{2\pi}} \exp\left(-\frac{1}{2z}\right), \tag{27}$$

where

$$z = \sum_{d=1}^{D} P/N^d. \tag{28}$$

We can now calculate the probability of a stable pattern, i.e., that the inequality $\xi_i^1 h_i > 0$ is satisfied for all $i$, with

$$\Pr(\text{stable pattern}) \approx \left[1 - \sqrt{\frac{z}{2\pi}} \exp\left(-\frac{1}{2z}\right)\right]^N$$
$$\approx 1 - N\sqrt{\frac{z}{2\pi}} \exp\left(-\frac{1}{2z}\right). \tag{29}$$

Since $N \to \infty$, Equation 29 will be close to 1 as the second term will be negligible. This will always be true if

$$z = \frac{1}{2\ln N}. \tag{30}$$

Therefore, since we have $N$ neurons with $\sum_{d=1}^{D} N^d$ connections between them, the maximum number of patterns we may store (while accepting small errors) is

$$p_c = \frac{\sum_{d=1}^{D} N^d}{2 \ln N}. \tag{31}$$

Or, if we cannot accept errors,

$$p_c = \frac{\sum_{d=1}^{D} N^d}{4 \ln N}. \tag{32}$$

$\square$

An additional informal perspective to consider regarding memory capacity is to notice that if $P$ scales with $N$, the ratio between the signal and noise terms will be constant. And, since $P \propto N^d$, the theoretical memory capacity scales polynomially with $N$ and linearly with $D$, and so the theoretical capacity is approximately $\sum_{d=1}^{D} c_d \cdot N^d$, where $c_d$ is a constant which depends on $d$.

### A.7 NUMERICAL IMPLEMENTATION OF SIMILARITY MEASURES

As in Millidge et al. (2022), in order to fairly compare similarity functions, we: (i) normalised similarity scores (separately for each similarity function) so their sum would be equal to 1 (since different measures had intrinsically different scales); and (ii) for distance measures, used the normalised reciprocal (since distance measures return low values for similar inputs, but the model relies on high values being returned for similar inputs, as in the dot product).

### A.8 SUPPLEMENTARY FIGURES AND TABLES

Table 3: Pearson correlation coefficients ($r$) between overlap and $\beta_1$ for mixed diluted networks from Table 2. Bolded values are significant at $\alpha = 0.05$ (without multiple comparisons adjustment). With multiple comparisons adjustment, there are no significant correlations. Given their construction, all networks have $\beta_0 = 1$, and although there is a small chance of 2–dimensional holes in some networks, we found that $\beta_{\geq 2} = 0$ for all simulated networks.

| No. patterns | 0.05N | 0.1N | 0.15N | 0.2N | 0.3N |
|---|---|---|---|---|---|
| R$\overline{12}$ | 0.02 | 0.09 | **0.21** | −0.15 | −0.01 |
| R1$\overline{2}$ | N/A | −0.05 | −0.1 | 0.01 | −0.08 |

Table 4: Extended list of network condition keys (top row), their number of non-zero weights for 1–, 2–, and 3–simplices (second, third, and fourth rows). $N$ is the number of neurons. For simulation, the number of simplices at each dimension are rounded to the nearest integer.

| | R3 | R$\overline{1}$23 | R1$\overline{2}$3 | R12$\overline{3}$ | R$\overline{123}$ |
|---|---|---|---|---|---|
| 1–simplices | 0 | $0.50\binom{N}{2}$ | $0.25\binom{N}{2}$ | $0.25\binom{N}{2}$ | $1/3\binom{N}{2}$ |
| 2–simplices | 0 | $0.25\binom{N}{2}$ | $0.50\binom{N}{2}$ | $0.25\binom{N}{2}$ | $1/3\binom{N}{2}$ |
| 3–simplices | $\binom{N}{2}$ | $0.25\binom{N}{2}$ | $0.25\binom{N}{2}$ | $0.50\binom{N}{2}$ | $1/3\binom{N}{2}$ |

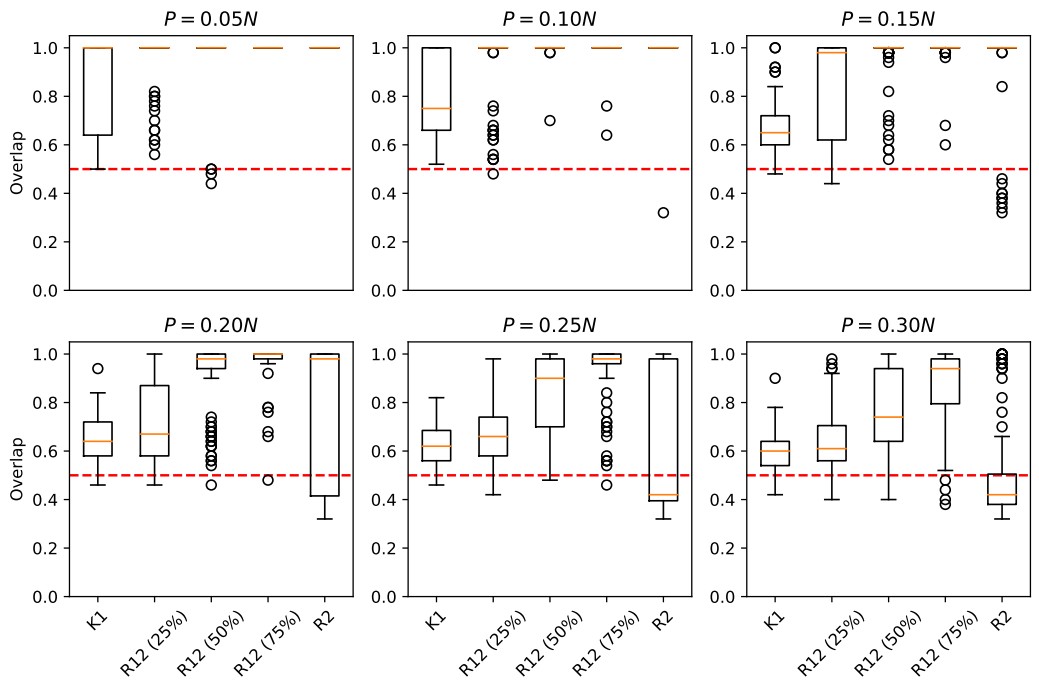

Figure 6: Box and whisker plots of final overlap distributions from traditional simplicial Hopfield networks with varying numbers of embedded patterns. Orange lines indicate the median. The red dashed line indicates an overlap of 0.5, chance.

Table 5: Mean $\pm$ standard deviation of overlap distributions ($n = 100$) from traditional simplicial Hopfield networks with varying numbers (top row) of random binary patterns. Keys per Table 4. At all pattern loadings, a one-way ANOVA showed significant variance between the networks ($p < 10^{-12}$, $F > 13.25$).

| No. patterns | $0.05N$ | $0.1N$ | $0.15N$ | $0.2N$ | $0.3N$ |
|---|---|---|---|---|---|
| R$\overline{1}$23 | $\mathbf{1 \pm 0}$ | $0.99 \pm 0.08$ | $0.97 \pm 0.17$ | $0.93 \pm 0.22$ | $0.89 \pm 0.15$ |
| R1$\overline{2}$3 | $\mathbf{1 \pm 0}$ | $\mathbf{1 \pm 0}$ | $0.98 \pm 0.05$ | $0.95 \pm 0.17$ | $0.91 \pm 0.18$ |
| R12$\overline{3}$ | $\mathbf{1 \pm 0}$ | $\mathbf{1 \pm 0}$ | $\mathbf{1 \pm 0}$ | $0.96 \pm 13$ | $0.93 \pm 0.13$ |
| $\mathbf{R\overline{123}}$ | $\mathbf{1 \pm 0}$ | $\mathbf{1 \pm 0}$ | $\mathbf{1 \pm 0}$ | $\mathbf{1 \pm 0}$ | $\mathbf{1 \pm 0}$ |
| R3 | $0.94 \pm 0.06$ | $0.78 \pm 0.14$ | $0.52 \pm 0.15$ | $0.51 \pm 0.13$ | $0.51 \pm 0.14$ |

Table 6: Mean $\pm$ standard deviation ($n = 10$) of fraction of correctly recalled MNIST memory patterns in simplicial Hopfield networks at a memory loading of 1000 memories. Network performance varied significantly.

| | K1 | R$\overline{1}$2 | R1$\overline{2}$ | $\mathbf{R\overline{123}}$ |
|---|---|---|---|---|
| Euclidean | $1 \pm 0$ | $1 \pm 0$ | $1 \pm 0$ | $\mathbf{1 \pm 0}$ |
| Manhattan | $1 \pm 0$ | $1 \pm 0$ | $1 \pm 0$ | $\mathbf{1 \pm 0}$ |
| Dot Product | $0.93 \pm 0.03$ | $0.93 \pm 0.02$ | $0.94 \pm 0.02$ | $\mathbf{1 \pm 0}$ |
| ced | - | $0.90 \pm 0.02$ | $0.91 \pm 0.03$ | $\mathbf{1 \pm 0}$ |
| cmd | - | $0.95 \pm 0.03$ | $0.97 \pm 0.03$ | $\mathbf{1 \pm 0}$ |

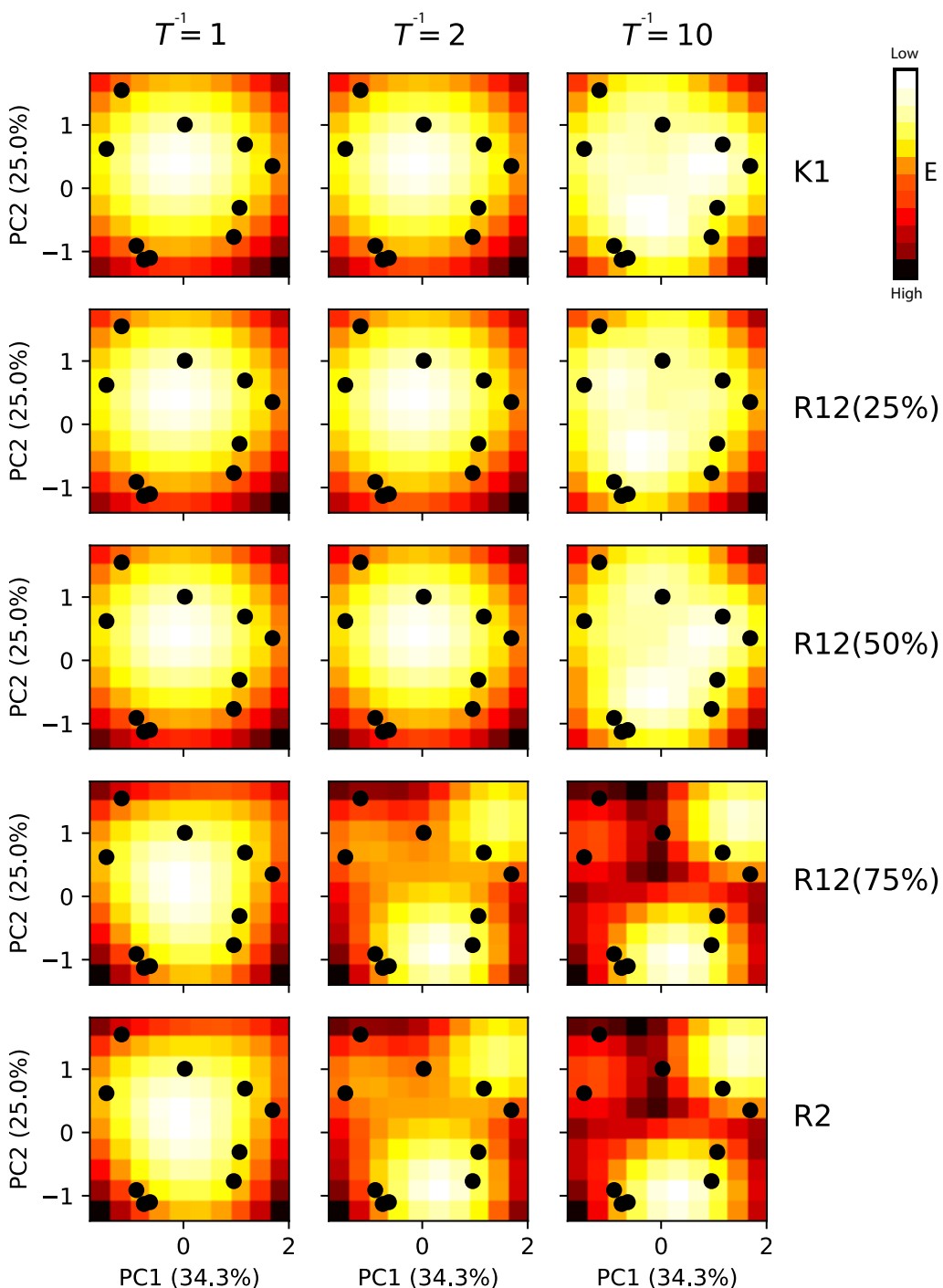

Figure 7: Relative energies of continuous modern networks in a $10 \times 10$ grid plane of the first two dimensions of PCA space, as computed using the memory patterns. Each network has $N = 10$ and the same $P = 10$ memory patterns are embedded. Network conditions vary by row and inverse temperatures vary by column. Black dots are projections of the 10 embedded patterns in the PCA space. The combined explained variance of the first two principle components is 59.3% of the memory patterns.

Table 7: Same as Table 6 but for CIFAR-10.

|  | K1 | R$\bar{1}$2 | R1$\bar{2}$ | **R$\bar{1}$2$\bar{3}$** |
|---|---|---|---|---|
| Euclidean | $0.31 \pm 0.08$ | $0.39 \pm 0.08$ | $0.51 \pm 0.07$ | **$0.64 \pm 0.08$** |
| Manhattan | $0.70 \pm 0.06$ | $0.77 \pm 0.06$ | $0.90 \pm 0.05$ | **$0.97 \pm 0.04$** |
| Dot Product | $0.50 \pm 0.07$ | $0.56 \pm 0.07$ | $0.68 \pm 0.06$ | **$0.72 \pm 0.08$** |
| ced | - | $0.61 \pm 0.06$ | $0.74 \pm 0.06$ | **$0.81 \pm 0.07$** |
| cmd | - | $0.65 \pm 0.07$ | $0.91 \pm 0.06$ | **$0.99 \pm 0.02$** |

Table 8: Same as Table 6 but for Tiny ImageNet.

|  | K1 | R$\bar{1}$2 | R1$\bar{2}$ | **R$\bar{1}$2$\bar{3}$** |
|---|---|---|---|---|
| Euclidean | $0.31 \pm 0.15$ | $0.34 \pm 0.14$ | $0.55 \pm 0.12$ | **$0.61 \pm 0.15$** |
| Manhattan | $0.63 \pm 0.10$ | $0.70 \pm 0.08$ | $0.84 \pm 0.08$ | **$0.91 \pm 0.09$** |
| Dot Product | $0 \pm 0$ | $0 \pm 0$ | $0 \pm 0$ | $0 \pm 0$ |
| ced | - | $0.51 \pm 0.14$ | $0.65 \pm 0.13$ | **$0.70 \pm 0.11$** |
| cmd | - | $0.71 \pm 0.08$ | $0.92 \pm 0.06$ | **$0.95 \pm 0.06$** |

