# OpenReview forum: "Simplicial Hopfield networks"
_ICLR.cc/2023/Conference — ICLR 2023 poster_

### Official Review · Reviewer_qaFD · 2022-10-22

**Confidence:** 3
**Correctness:** 3
**Technical Novelty And Significance:** 3
**Empirical Novelty And Significance:** 3
**Recommendation:** 8

**Clarity, Quality, Novelty And Reproducibility:**

To my knowledge the work is original in that it is the first to characterize the memory capacity of Hopfield networks beyond pairwise only connections. The writing is clear, though Figure 1 doesn’t help explain the idea well.

**Strength And Weaknesses:**

******************Strengths******************

- (+ +) The idea of treating Hopfield Networks as simplicial complexes is enticing and indeed novel.
- (+) The experimental results are minimal and simple, but still convincing. The memory capacity of a Hopfield Net, in the binary case, is noticeably improved by including higher degree connections.

********************Weaknesses********************

- (- -) The experimental results are minimal and simple. Only K=1 and K=2 simplices are considered (see below). In particular, the continuous case chooses only one “simplicial model” and applies it on MNIST, CIFAR, and Tiny ImageNet while comparing different similarity metrics. Comments:
    - It would be useful to see behavior across simplicial models in the continuous case. In particular, perhaps, simplicial networks of higher order than just 2.
    - Figure 2 describing these results is laid out such that it is very difficult to compare the baseline to the proposed model… This is fine, but perhaps a corresponding table like Table 2 for recall on the different datasets for one of the distance metrics would be preferable.
- (-) The re-formulation of the Hopfield Network using simplices is convenient for mathematical proofs, but practically it can be computationally prohibitive to consider all the subsets that higher degree simplicial complexes create. This is shown by the experiments which exclusively considered pairwise and K=2 degree complexes. To sell the practical application of this idea, I encourage the authors to include experiments with higher order simplicial complexes, or to reconsider use of the hypergraph formulations as discussed in the Appendix.
- (-) I found Figure 1 completely unhelpful in understanding the paper. 1B has no meaning to me, only that the weight distributions “look” the same at different simplices (there’s not even a y-axis). 1C is not sufficiently explained and is equally meaningless to me.

**Additional comments**

- (-) Please revise the statement “Ramsauer et al. (2021) extended modern Hopfield networks to allow neurons to take continuous values – where previously they took only binary values”. Ramsauer et al. claimed to have allowed Hopfield networks to take continuous states, but in fact continuous Hopfield networks were introduced by [Hopfield himself in 1984](https://www.pnas.org/doi/10.1073/pnas.81.10.3088). We want to correct this incorrect claim as much as possible in papers moving forward.
- (-) There is no mention of code to be released for reproducibility.
- (-) Notation in equation (5) is strange. Why input “$\ldots \Xi^T S \ldots$” to the logsumexp when the RHS uses “$\ldots \xi_\sigma^\mu S_\sigma \ldots$”?
- (-) Where does $M^2$ come from in equation (6)? This is seemingly copied directly from Ramsauer et al. and not defined here.

**Questions**

- The original HN uses a dot product, which is clearly a similarity measure where more similar inputs have a higher value. However, taking the Manhattan or Euclidean distance is a **********difference********** measure where more similar inputs have a smaller value. Is the inverse of the euclidean distance taken in the experiments? Please describe this for reproducibility.

**Summary Of The Paper:**

The authors introduce Simplicial Hopfield Networks — an architecture that extends the original Hopfield Network by allowing setwise connections in addition to pairwise connections between all neurons in the network. This reformulation allows higher memory capacity and performance and significantly increases the memory capacity of the traditional Hopfield Network.

**Summary Of The Review:**

The paper offers a novel perspective into how to interpret Hopfield Networks, though reproducibility of this paper is difficult in its current state (see weaknesses). I recommend this as borderline accept, with room for improvement if my comments can be addressed.

I did not read the appendix in depth.

---

> ### Author Response · Authors · 2022-11-15
> **Response to Reviewer qaFD**
>
> Thank you for your very insightful review and suggestions.
>
> - In response to your suggestions regarding higher-order simplicial models, we have run further testing on binary and continuous models with $\text{dim}(K)=3$. These results are shown in Appendix A8 of the revised paper.
> - We also agree that, for clarity and ease of comparison, a tabular form of the results from Figure 2 is helpful for readers. As you suggested, we have added such tables (based on Table 2) in Appendix A8.
>
> Figure 1B: This panel shows the weights, at different dimensions, of a simplicial Hopfield network. The weights come from Equation 1, where the term $1/N$ normalises the values from $[-P,+P]$ to $[-P/N,+P/N]$. In particular, this example uses $N=100$ and $P=10$, so the x-axis shows values ranging from $[-0.1,+0.1]$. An explanation about this has been added to the figure caption for clarity. The y-axes (now labelled in the revision) are the relative frequencies of each weight value bin in the histogram. We chose to plot the relative frequencies because in this case there are far more higher-dimensional connections than lowerer-dimensional connections. The reason these panels all look similar is because, in the binary pure simplicial complex case, they all stem from the same underlying 'random walk' processes, just at different dimensions.
>
> Figure 1C: This panel attempts to illustrate two aspects of simplicial Hopfield networks: (i) the hierarchical relationship between elements in the simplicial complex (in this example, up to $3$--simplices), particularly the relationship between (co)faces in their geometrical realisations (points, lines, triangles, and tetrahedra); and (ii) how such a structure could permit certain weight modulation or interaction between (co)faces, or modulation/interaction within the same dimension (e.g., using Hodge Laplacians, which are discussed in Appendix A3). Biological interpretations of such interactions/modulations are discussed in Appendices A2 and A3. We have added some further explanations to the legend for Figure 1C and to Appendices A2 and A3 to help explain these ideas. If it is still unclear, we are happy to explain further. Alternatively,, if you or others recommend it, we could remove this panel or move it to the appendix as a separate figure, since we didn’t actually utilise such interactions/modulations in the experiments.
>
> Thank you for correcting us regarding the development of continuous Hopfield networks, which, as you say, were originally introduced by Hopfield (1984). Reviewer R7jA also kindly pointed out that Krotov & Hopfield (2016) introduced continuous modern Hopfield networks, not Ramsauer et al. 2021. The relevant section of the introduction now reads: “Krotov & Hopfield (2016) (like Hopfield (1984)) also investigated neurons which took on continuous states. Upon generalising this model by using the softmax activation function, Ramsauer et al. (2021) showed a connection to the attention mechanism of Transformers (Vaswani et al. (2017)).” Please let us know if there are any other missing references or oversights regarding this literature which you are aware of. We certainly don’t wish to misrepresent any claims or past work.
>
> Thank you for reminding us regarding reproducibility. We also noticed the ICLR Author Guide (https://iclr.cc/Conferences/2022/AuthorGuide) suggests including a Reproducibility Statement at the end of the main text. We have now added this statement and will be uploading our code as supplementary material.
>
> Your question regarding notation in Equation 5 was actually picking up on a typo. Thanks for spotting it. We have now corrected it to $\Xi$ on the RHS.
>
> The terms $T\text{log}N + (1/2) M^2$ from Equation 6 are normalisation constants to bound the energy between $0$ and $2M^2$, where $M$ is the Euclidean norm of the largest pattern vector. Indeed, these terms are equivalent to those found in Ramsauer et al. (2021). Reviewer R7jA pointed out that since they are constants, they can safely be dropped; the spin update rule can be obtained from just the $S$-dependent terms in $E$, and those last terms do not have a dependence on $S$. We have therefore removed these terms from Equation 6 in the revised version of the manuscript.
>
> Indeed, distance measures such as the Manhattan and Euclidean distance return low values for similar inputs, whereas the dot product returns high values. As in Millidge et al. (2022), for distance measures we use their reciprocal. Additionally, in order to fairly compare similarity functions, we normalised similarity scores (separately for each similarity function) so their sum is equal to 1 (since different measures have intrinsically different scales). As you suggested, we have added this information in the appendix for clarity and to aid reproducibility.
>
> Your questions, feedback, and suggestions are very helpful. Thank you very much for your efforts in helping us improve the paper.

---

> > ### Comment · Reviewer_qaFD · 2022-12-12
> > **A welcomed improvement over the first submission**
> >
> > I thank the authors for making my requested adjustments to the manuscript. I have increased my score accordingly.

---

### Official Review · Reviewer_fyDU · 2022-10-23

**Confidence:** 3
**Clarity, Quality, Novelty And Reproducibility:** This paper is of good quality and the…
**Correctness:** 3
**Technical Novelty And Significance:** 3
**Empirical Novelty And Significance:** 3
**Recommendation:** 8

**Strength And Weaknesses:**

The paper is clearly written. Both theoretical analysis and empirical simulations are carried out to prove their ideas. My main concern is that, why the capacity in Hopfield nets is a problem, especially in the biological brain where there are a large number of neurons? As long as there are enough neurons, the memory capacity is not a problem (we don't need to memorize many items at the same time). Is it more about a mathmetical "game" or has some real-world implications?

**Summary Of The Paper:**

The authors introduced simplical complexes into various Hopfield networks and showed that the memory capacity of the modified Hopfield nets are increased. They shoed that such improvement comes from the topology rather tha the additional information in the form of parameters. They also whoed that how distance measures of a more "geometric flavour" can further improve performance in these networks.

**Summary Of The Review:**

The study is comprehensive, and both theoretical analysis and simulation results are carried out to prove their ideas.

---

> ### Author Response · Authors · 2022-11-15
> **Response to Reviewer fyDU**
>
> Thank you very much for your review.
>
> Your questions prompted us to consider some rather detailed and broad issues. For the benefit of other readers we have chosen to add these responses in the form of an additional appendix (A1) in the revised manuscript. There we discuss these issues from computational, biological, and psychological perspectives. More directly, and in summary: it is partly a theoretical/computational game to improve memory capacity of these models (which offers potential advantages and future directions for machine learning researchers, especially those working with Transformers), and it is partly a theoretical finding which has implications for brain function (which may guide future experimental and theoretical/computational research in areas such as dendritic integration, glia-neuron computation, and many other areas). We would very much welcome your feedback on this new appendix, especially if you believe there are relevant papers or evidence which can be added.
>
> We appreciate your time and effort in helping us improve the manuscript.

---

### Official Review · Reviewer_R7jA · 2022-10-23

**Confidence:** 4
**Correctness:** 4
**Technical Novelty And Significance:** 3
**Empirical Novelty And Significance:** 3
**Recommendation:** 8

**Clarity, Quality, Novelty And Reproducibility:**

Potentially novel and interesting paper, but requires clarifications in the presentation.

**Strength And Weaknesses:**

This paper connects many interesting ideas, such as Hopfield networks, simplicial complexes, many-body synapses in biology, etc. These are great topics, and it is nice to see them discussed in the paper. However, I am struggling understanding the core mathematical definitions of the proposed model. Specifically,

1. I do not understand the definition and notations of the model (equation 1). Could the authors please write down (possibly in the appendix) the complete expression for the energy function in some toy model example (say 2 memory patterns, upto 3-simplex connections). I want to see the complete expression for the energy function written in terms of $S_i$ and $\xi^\mu_i$.

2. Is the energy function (equation 1) a sum of the energy functions of Krotov & Hopfield 2016 with say 3-neuron connections, 2-neuron connections, 1-neuron connections? If so, are the weights of those connections independent of each other for 3-, 2- and 1- body interactions? In the model of Krotov & Hopfield, once the patterns $\xi^\mu_i$ are fixed (consider a power model for simplicity with n=3), the connectivity - symmetric tensor of the 3rd rank $w_{ijk}$ - is completely determined. This tensor has $N(N-1)(N-2)/6$ elements in it, but those elements are not independent. Are the authors proposing to declare some of the elements of this tensor to be zero, so that only those terms present in the simplicial complex are non-zero?

3. Also, I don’t fully understand the definitions of the models considered in Table 1. For the conventional 2-body Hopfield network if the number of patterns is odd, then the connectivity matrix $w_{ij}$ has all non-zero entries. There is not any flexibility to delete any entries in it. How can the authors decrease the number of parameters from $N(N-1)/2$ to $0.75N(N-1)/2$? Which elements of the matrix $w_{ij}$ are deleted? Similarly, how do authors add 3-body connections?

Additionally, I do not understand the histograms in Figure 1B. If the memories in the Hopfield network are binary ($\pm 1$) patterns and there are P of those, the elements of the weight matrix belong to the range $-P$ to $+P$ and are all integers. How do the authors get numbers smaller than 1 on the x-axis? Also, these three histograms look very similar to me. Is this what the authors are trying to illustrate by showing them? I somehow could not find a clear conclusion from these histograms in the text.

Continuous modern Hopfield networks were introduced in Krotov & Hopfield 2016, not in Ramsauer et al. 2021. Ramsauer et al. 2021 generalized the model of Krotov and Hopfield (which used any rapidly growing neuron-wise activation function, e.g. power function) to the case of softmax activation function, and made the connection with transformers. However, the theoretical framework for continuous modern Hopfield networks has already been introduced in Krotov & Hopfield 2016, see for example equation (10) there, and numerical experiments with MNIST (all use continuous variables).

While I greatly appreciate that the authors discuss legitimate biophysical mechanisms for building set-wise connections in the brain (Appendix A1), these set-wise connections can also be built from simple pair-wise connections by introducing additional hidden neurons, see for example https://openreview.net/forum?id=X4y_10OX-hX. This possibility should at least be mentioned somewhere in the paper. Even if one does not use more rarely discussed biophysical mechanisms mentioned by the authors in Appendix A1, there is nothing unbiological (except for shared weights) in many-body set-wise connections.

Typo: the constant factor $T$ should be replaced by $1/T$ in equation (7).

A small suggestions to the authors for making their formulas more compact (equations 5, 6): $\frac{1}{T^{-1}} = T$. Also, the last two terms in equation (6) are just constants, thus can be dropped.

The notation $\dot{S}$ in equations 2, 4, etc., is somewhat confusing, since it is often used for denoting the time derivative. While here, the authors use it for denoting the updated state of the network. I suggest that the authors either replace it by something else, or add a note the first time it appears in the paper about its meaning.


**Summary Of The Paper:**

This paper proposes to add set-wise connections to pair-wise connections in the Hopfield networks. The main contribution of the paper is the claim that adding such connection improves the memory storage properties of these networks. Interesting biological mechanisms of building many-body synapses are also mentioned.

**Summary Of The Review:**

Potentially interesting paper, but requires clarifications regarding the core mathematical definitions and more accurate description of the relationship to previous work. For now, I will score this paper in the middle range. I am keeping an open mind about this work, and depending on the revisions/discussions with the authors can either increase or decrease the scores.

---

> ### Author Response · Authors · 2022-11-15
> **Response to Reviewer R7jA**
>
> Thank you for your very helpful and diligent review.
>
> 1. We have added a worked example to the paper (in Appendix A4 of the latest version) for a 3-complex Hopfield network on $N=6$ neurons with $P=3$ embedded memory patterns. We are more than willing to expand upon this example or add additional examples if you think this would be helpful. Please let us know, as we hope to make the paper and ideas as accessible as possible.
> 2. Yes, the energy function (for the pure simplicial k-complex case in a modern network) is essentially a sum of the energy functions of Krotov & Hopfield (2016) with all possible k-connections, but where the weights of those connections are independent of each other for each level of interaction. However, such networks possess many parameters, which is why we were interested in the diluted case. In the diluted case, a subset of weights at each dimension are randomly selected and set to 0, thus 'functionally deleted'. In these cases, we limit the total number of parameters used to store memories in the network and test whether 'trading' some connections of particular orders, e.g., 2-body connections for 3-body connections, benefits performance (without increasing the overall number of connections) – and we show it can.
> 3. We randomly delete connections (in practice, we normally compute only those which are randomly chosen to remain in the network, rather than computing things we will then throw away). This leads to, we think, a quite remarkable result: without any form of cleverness or optimization, but by just blindly ignoring some connections, e.g., 2-body, and replacing them with connections of a higher order (again blindly), e.g. 3-body, we can see marked performance improvements.
>
> Figure 1B: The weights come from Equation 1, where the term $1/N$ normalises the values from $[-P,+P]$ to $[-P/N,+P/N]$. In particular, this example uses $N=100$ and $P=10$, so the x-axis shows values ranging from $[-0.1,+0.1]$. An explanation about this has been added to the figure caption for clarity. The y-axes (now labelled in the revision) are the relative frequencies of the weights. We chose to plot the relative frequencies because in this case there are far more higher-dimensional connections than lowerer-dimensional connections, so plotting the absolute frequencies would make them more difficult to compare. The reason these panels all look similar is because, in the binary pure simplicial complex case, they all stem from the same underlying 'random walk' processes, just at different dimensions.
>
> Thank you for correcting us regarding the development of continuous modern Hopfield networks, which indeed were introduced in Krotov & Hopfield (2016), not in Ramsauer et al. (2021). We also note that continuous traditional (non-modern) Hopfield networks were also introduced by Hopfield himself in Hopfield (1984), as noted by Reviewer qaFD. The relevant section of the introduction now reads: “Krotov & Hopfield (2016) (like Hopfield (1984)) also investigated neurons which took on continuous states. Upon generalising this model by using the softmax activation function, Ramsauer et al. (2021) showed a connection to the attention mechanism of Transformers (Vaswani et al. (2017)).” Please let us know if there are other missing references or oversights regarding this literature which you are aware of.
>
> We also thank you for referring us to Krotov & Hopfield (2021), which indeed shows how setwise connections can also be built from simple pairwise connections by introducing additional hidden neurons. We have included mentions of this in the main text and in Appendix A2.
>
> Thank you, also, for your notation suggestions. In response, we
> - fixed the typo you found in Equation 7;
> - replaced $\dfrac{1}{T^{-1}}$ with $T$ for compactness in Equation 5;
> - dropped the last two terms ($T\text{log}N + (1/2) M^2$) from Equation 6, since the spin update rule can be obtained from just the $S$-dependent terms in $E$, and those last terms do not have a dependence on $S$; and
> - replaced the $\dot{S}$ notation (which we agree is non-conventional and likely to confuse some readers) with time-step notations ($S^{(t)}$) throughout the main text and appendices to denote spin states and updates.
>
> Thank you again for your time and helpful feedback.

---

> > ### Comment · Reviewer_R7jA · 2022-12-09
> > **Thank you for the clarifications**
> >
> > Thank you for answering my questions. This is a nice paper, which adds valuable perspective to the existing literature on Hopfield networks. My concerns have been addressed and I support acceptance.

---

### Decision · Program_Chairs · 2023-01-20

**Decision:**

Accept: poster

**Justification For Why Not Higher Score:**

All reviewers recommend accept, but there were not advocates for a higher rating.

**Justification For Why Not Lower Score:**

All agree the article should be accepted.

**Metareview: Summary, Strengths And Weaknesses:**

The article studies Hopfield networks with added set wise connections in regard to the number of stable attracting memory patterns.

* Strengths are the extension of Hopfield networks to set wise connections, discussion of higher memory capacity, and empirical studies.
* Weaknesses are concerns about the simplicity of experiments, practicality of the reformulation of the Hopfield networks, and limited reproducibility.

During the discussion period some concerns from the initial reviews could be resolved and authors added details to appendix, which prompted some of the reviewers to raise their scores. Reviewers mostly agree that the article is of good quality and well supported claims of clear logic. I conclude the article makes interesting, novel, and sufficiently well-developed contributions. In view of the favorable and in parts enthusiastic feedback from the reviewers I am recommending accept. The authors are encouraged to upload their code as supplementary material as promised in their responses.

**Note From Pc:**

if the above contains the word "oral" or "spotlight" please see: "oral" presentation means -> notable-top-5% and "spotlight" means -> notable-top-25%. As stated in our emails, we are disassociating presentation type from AC recommendations

**Summary Of Ac-Reviewer Meeting:**

NA